# A class of secreted mammalian peptides with potential to expand cell-cell communication

Amanda L. Wiggenhorn [1,2,3,4], Hind Z. Abuzaid[1,3], Laetitia Coassolo[1], Veronica L. Li[1,2,3,4], Julia T. Tanzo[1,3], Wei Wei [1,3,5], Xuchao Lyu [1,3,4], Katrin J. Svensson [1,6] & Jonathan Z. Long [1,2,3,4,6] ✉

Peptide hormones and neuropeptides are signaling molecules that control diverse aspects of mammalian homeostasis and physiology. Here we provide evidence for the endogenous presence of a sequence diverse class of blood-borne peptides that we call "capped peptides." Capped peptides are fragments of secreted proteins and defined by the presence of two post-translational modifications – N-terminal pyroglutamylation and C-terminal amidation – which function as chemical "caps" of the intervening sequence. Capped peptides share many regulatory characteristics in common with that of other signaling peptides, including dynamic physiologic regulation. One capped peptide, CAP-TAC1, is a tachykinin neuropeptide-like molecule and a nanomolar agonist of mammalian tachykinin receptors. A second capped peptide, CAP-GDF15, is a 12-mer peptide cleaved from the prepropeptide region of full-length GDF15 that, like the canonical GDF15 hormone, also reduces food intake and body weight. Capped peptides are a potentially large class of signaling molecules with potential to broadly regulate cell-cell communication in mammalian physiology.

Peptide hormones and neuropeptides are fundamental signaling molecules that mediate cell-cell communication[1]. These signaling molecules are produced by proteolytic cleavage of secreted pre-proproteins and released extracellularly via the classical secretory pathway. Once secreted, peptide hormones and neuropeptides act on cognate receptors to regulate nearly all aspects of homeostasis and physiology. Because of their potent and powerful physiologic actions, peptide hormones and neuropeptides have attracted considerable pharmaceutical interest as starting points for the development of therapeutics across multiple human disease areas[2–4].

From a chemical perspective, a subset of mammalian neuropeptides/peptide hormones are unusual in that they contain co-incident N-terminal pyroglutamyl and C-terminal amide post-translational modifications. Representative examples include TRH (pGlu-HP-NH2) and GnRH (pGlu-HWSYGLRPG-NH2). These co-incident terminal modifications are installed via the action of two enzymes, PAM and GC, and function to enhance peptide signaling

and bioactivity[5,6]. For instance, removal of both terminal modifications of TRH renders the resulting unmodified peptide devoid of agonist activity at the TRH receptor and highly sensitive to proteolytic degradation[7]. Beyond this subset of neuropeptides and peptide hormones, co-incident N-pyroglutamyl/C-amide modifications have not been identified, suggesting that they appear to be restricted to and designate a subset of privileged sequences that encode for bioactive signaling peptides.

We hypothesized that such peculiar and co-incident N-pyroglutamyl/C-amidation modifications of peptides are not installed by happenstance, but instead define a chemical motif that designates more potentially bioactive signaling peptides than has been reported to date. This hypothesis was inspired by the well-established observation that certain chemical motifs already define classes of molecules and functions. For instance, a free amino group is characteristic of monoamines; a cyclized arachidonate acid is characteristic of prostanoids, and a cholesterol backbone is characteristic of steroids.

[1]Department of Pathology, Stanford University School of Medicine, Stanford, CA, USA. [2]Department of Chemistry, Stanford University, Stanford, CA, USA. [3]Sarafan ChEM-H, Stanford University, Stanford, CA, USA. [4]Wu Tsai Human Performance Alliance, Stanford University, Stanford, CA, USA. [5]Department of Biology, Stanford University, Stanford, CA, USA. [6]Stanford Diabetes Research Center, Stanford University, Stanford, CA, USA. ✉e-mail: jzlong@stanford.edu

Here, we provide experimental evidence for the endogenous presence of a number of additional peptides with co-incident N-pyroglutamyl/C-amidation modifications in both mouse and human plasma by combining a computational prediction strategy with targeted mass spectrometry. We call these peptides "capped peptides" due to these chemical modifications which function as terminal caps of the intervening peptide sequence. Capped peptides also exhibit regulatory characteristics similar to other signaling peptides, including dynamic circulating levels in response to physiologic and environmental state. In vitro and in vivo functional assays establish signaling and bioactivity for two capped peptides: CAP-TAC1 is a tachykinin neuropeptide-like molecule that exhibits nanomolar agonist activity at multiple mammalian tachykinin receptors, and CAP-GDF15, derived from the prepropeptide region of the anorexigenic hormone GDF15, is itself a 12-mer anorexigenic peptide. Our studies demonstrate that N- and C-terminal capping chemical motif that is present in more endogenous secreted peptides than previously reported. Capped peptides therefore constitute a class secreted signaling peptides with potential to broadly regulate cell-cell communication in mammalian physiology.

## Results

### Genomic and empirical evidence for capped peptides in mouse plasma

To define potential N-pyroglutamyl/C-amide modified sequences from existing classically secreted proteins, we first used Uniprot[8] to curate a collection of protein sequences corresponding to classically secreted mouse proteins (Methods). Our initial collection of $N = 2835$ sequences contained many known classically secreted proteins, including apolipoproteins, secreted enzymes, and preprohormones (Supplementary Data 1). C-terminal amidation sequences were defined by the presence of a glycine-dibasic GKR/GRR tripeptide (Fig. 1a). N-terminal pyroglutamylation sequences were identified by the presence of a glutamine (Q) upstream of the glycine-dibasic motif (Fig. 1a, Supplementary Data 2). By length, we restricted our search criteria to those peptides 20 amino acids or shorter because this is the upper limit for reliable chemical synthesis of authentic peptide standards; in addition, multiple known peptides with capped characteristics are shorter than this length.

Using this computational framework, we predicted a total of 216 potential cleaved and modified mouse peptides from 186 classically secreted proteins encoded in the mouse genome (Fig. 1b). To determine whether capped peptides are produced endogenously, we used targeted liquid chromatography/mass spectrometry (LC-MS) approach to directly measure the levels of all predicted capped peptides in mouse plasma. Importantly, this targeted mass spectrometry-based workflow obviates the need for identification via database searching that is classically associated with untargeted peptidomics or shotgun proteomics. Our two-step mass spectrometry procedure involved first identifying endogenous peaks with the same MS1 mass-to-charge (m/z) ratio and retention time as our authentic standard using a high-resolution time-of-flight instrument (LC-QTOF); next, we developed multiple reaction monitoring methods on a triple quadrupole instrument (LC-QQQ) that enabled isolation, fragmentation and detection of a specific parent-to-daughter transition characteristic of each authentic peptide standard (see Methods for mass spectrometry details and Fig. 1c).

Experimentally, total mouse plasma peptidome was isolated using previously described protocols (see Methods). In parallel, we used solid-phase peptide synthesis to generate authentic peptide standards for all 216 predicted capped peptides (Fig. 1c). Both mouse plasma peptide and the mixture of the 216 peptide standards were separately reduced with DTT, alkylated with iodoacetamide, and concentrated using C8 columns. In the first step, we identified 61 peptides had an MS1 peak that exhibited identical retention times (within 1 minute) and m/z ratio (within 20 ppm) as the authentic standard (Fig. 1d). Next, we developed multiple reaction monitoring (MRM) methods on an LC-QQQ that monitored for each peptide for the presence of a specific parent-to-daughter transition characteristic of the authentic standard for these 61 peptides. Using this MRM protocol, we validated the endogenous presence of transitions corresponding to 39 peptides (Fig. 1d, Supplementary Data 2, and Supplementary Fig. 1). By comparison to an external standard curve, the mouse capped peptides exhibited circulating concentrations in the range of ~0.1–100 nM (Fig. 1e).

A representative example of a positive detection event for CAP-TAC1 (pGlu-FFGLM-NH2, Fig. 1f), a capped peptide derived from amino acids 63–68 of full-length TAC1 (protachykinin-1), is shown in Fig. 1g, h. Here, the authentic CAP-TAC1 standard exhibits identical m/z ratio and retention time to an endogenous plasma peak (m/z = 724.3, retention time = 27 min, Fig. 1g). In optimizing the MRM method using an authentic CAP-TAC1 standard, we identified a characteristic and prominent 724.4 > 463.2 transition which corresponded to the b4 daughter ion (Supplementary Data 2 and Supplementary Fig. 1). As shown in Fig. 1h, by LC-QQQ we also successfully detected an endogenous plasma peak with transition 724.4 > 263.2 eluting at the same time as the authentic standard.

To exclude the possibility that N-pyroglutamylation may be artefactually occurring in the sample preparation, we subjected a synthetic standard of uncapped CAP-TAC1 (QFFGLM) to the sample preparation conditions. As shown in Supplementary Fig. 2A, we did not observe any formation of pGlu-FFGLM from the QFFGLM starting material; in addition, pGlu-FFGLM exhibited a distinct retention time in comparison to CAP-TAC1 (Supplementary Fig. 2B). Changing the prediction criteria for the N-terminus to other, non-Q amino acids also produced ~50–300 predicted peptides (Supplementary Fig. 2C). The capped peptide sequences identified here are not found in PeptideAtlas, which may be attributable to the higher sensitivity, targeted mass spectrometry approach used here compared to shotgun approaches[9]. Lastly, we successfully acquired and were able to manually annotate full MS/MS spectra for CAP-TAC1 (Fig. 1i) as well as an additional 8 mouse capped peptides (Supplementary Fig. 3), providing further experimental evidence for their endogenous presence in mouse plasma.

These data provide mass spectrometry evidence that capped peptides are endogenously circulating, blood-borne molecules. In addition, we establish that specific proteolytic processing and capping to produce protected peptide fragments is much more prevalent than previously anticipated, and certainly extends beyond the known subset of neuropeptides and peptide hormones containing these modifications. Our inability to detect the full set of predicted capped peptides may either reflect true absence of post-translational processing to generate those fragments, circulating that are below our limit of detection, or local and organ-specific expression.

### Sequence and gene-level analysis of mouse capped peptides

We next examined the sequences and set of full-length pre-proprecursor proteins of the capped peptides for which we had mass spectrometry evidence for their endogenous presence. Two of the capped peptides, CAP-GNRH1 (pGlu-HWSYGLRPG-NH2) and CAP-GAST (pGlu-RPRMEEEEEAYGWMDF-NH2) directly corresponded with the known hormone sequences for GnRH and gastrin (Fig. 2a, b). These data demonstrate that our hybrid computational-analytical approach can "re-discover" two of the known signaling peptides that harbor both N-terminal pyroglutamylation and C-terminal amidation. An additional 12 capped peptides mapped to preproprecursor proteins corresponding to genes that had previously been annotated to generate polypeptides with signaling bioactivity, but for which a shorter cleavage fragment had not been previously identified. For instance, we observed a capped tripeptide, CAP-FGF5 (pGlu-WSPS-NH2), derived from amino acids 76–79 of the prepro-FGF5 (fibroblast growth factor

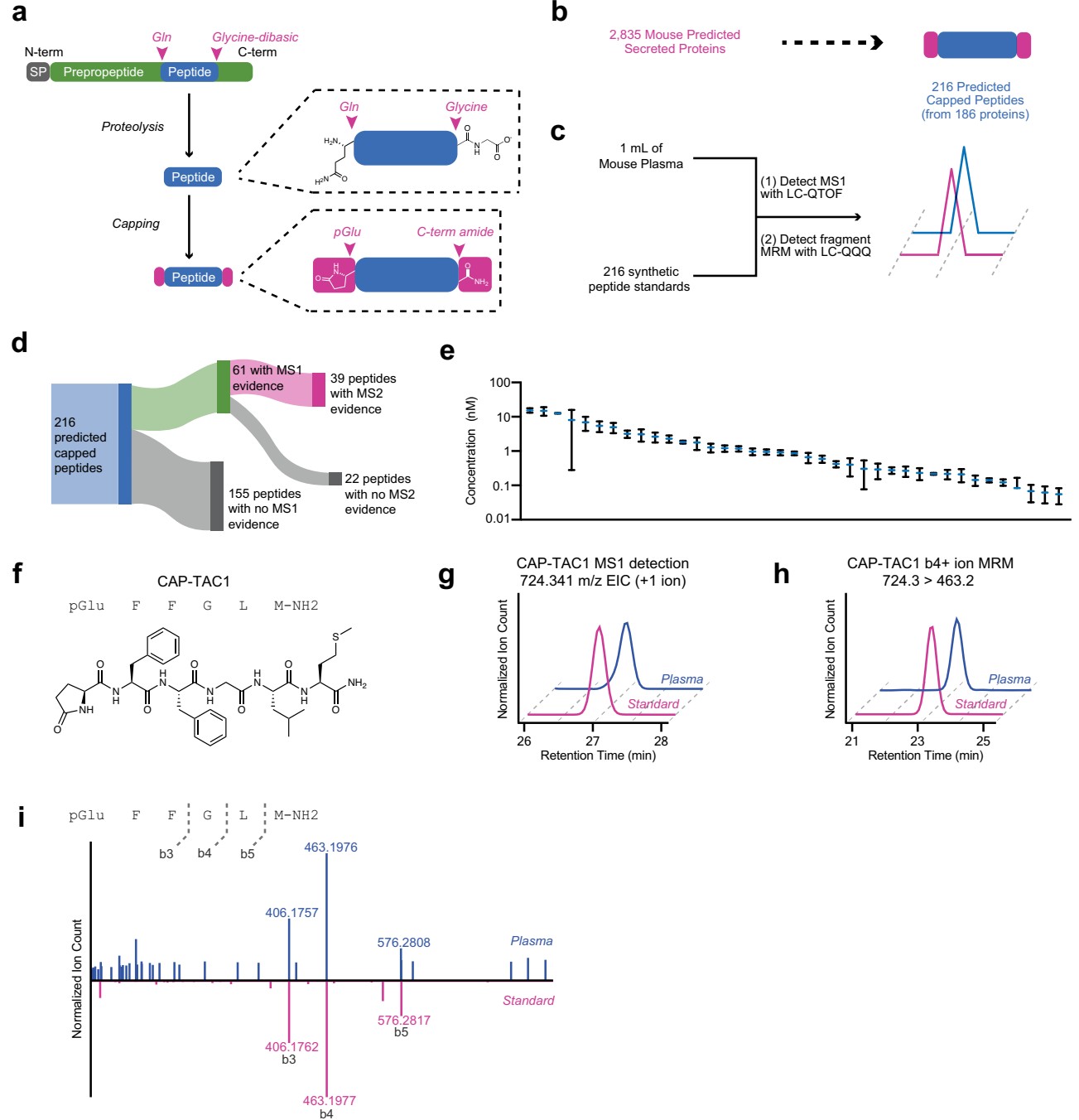

**Fig. 1 | Genomic prediction and mass spectrometry detection of capped peptides in mouse plasma. a** Schematic representation of capped peptide production from secreted, full-length preprecursor proteins. N-term, N-terminal; C-term, C-terminal; SP, signal peptide; Gln, glutamine; pGlu, pyroglutamyl. **b, c** Number of predicted mouse capped peptides (**b**) and the targeted mass spectrometry strategy for their detection (**c**). **d** Schematic representation of proportion of capped peptides with MS1 parent ion evidence (LC-QTOF) and parent-to-daughter transition (LC-QQQ) evidence. **e** Concentration distributions of detected capped. **f** Chemical structure of CAP-TAC1 (pGlu-FFGLM-NH2). **g–i** Representative extracted ion chromatogram of the parent +1 ion at m/z = 724.341 (20 ppm) (**g**) and the MRM 724.3 to 463.2 b4+ ion transition (**h**), and MS/MS mirror plot **i** of authentic CAP-TAC1 standard (pink) and the endogenous mouse plasma peak (blue). **e** Data are shown as means ± SEM of N = 3 biologically independent pooled mouse plasma samples. **g** Experiment was repeated three times; for **h, i**, experiment was completed once. Source data are provided as a Source Data file.

5) (Fig. 2b). A second capped peptide, CAP-GDNF (pGlu-AAAAS-PENSRGK-NH2), mapped to amino acids 94-106 of GDNF (glial cell line-derived neurotrophic factor) (Fig. 2b). Classically, FGF5 is a so-called "paracrine" member of the fibroblast growth factor (FGF) family and has diverse roles, including in the regulation of hair cycle and length[10]. GDNF is a major growth factor that promotes the survival of dopaminergic and motor neurons; outside of the nervous system, GDNF is also a morphogen in the kidney and a spermatogonia differentiation factor[11]. Our data suggest that FGF5 and GDNF might also exhibit endocrine functions via cleavage fragments generated from their canonical polypeptide sequences. Lastly, the remaining capped peptides (25/39, ~64%) mapped to preproprecursors for proteins and genes that had not been previously suggested to have any roles in signaling. These include CAP-COL27A1 (pGlu-LGPP-NH2), which is

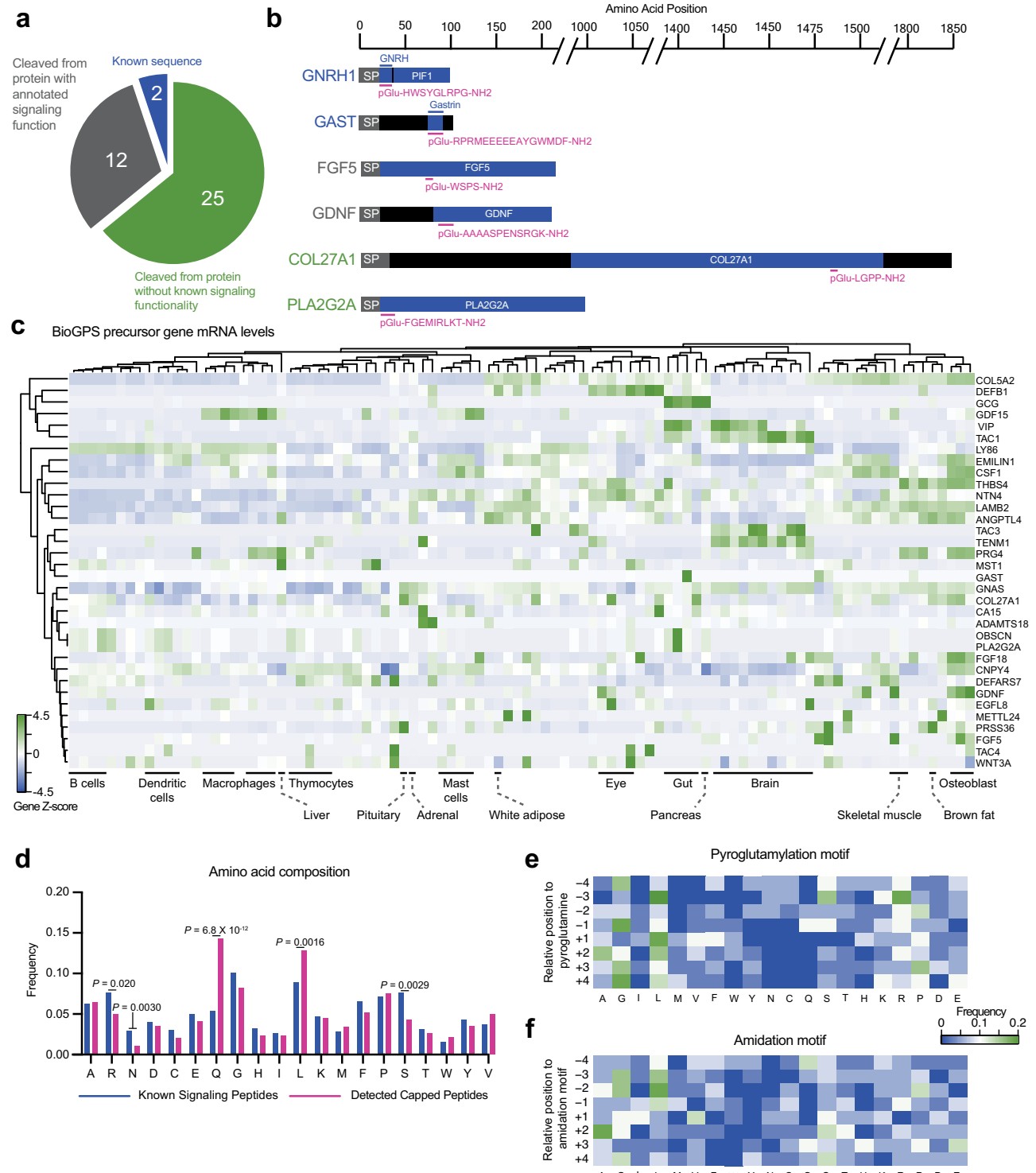

**Fig. 2 | Sequence and and gene-level analysis of capped peptides. a, b** Pie chart (**a**) and representative examples (**b**) of the distribution of mouse capped peptides based on their full-length preproprotein precursors. SP, signal peptide. **c** H-clustered heatmap of mRNA expression for capped peptide preproprecursor home genes across mouse tissues and cell types. **d** Frequency of each amino acid in capped peptides versus a Uniprot reference set of known peptide hormones and neuropeptides. **e, f** Heatmap of amino acid frequency within four residues upstream and downstream of the N-terminal pyroglutamylation (**e**) or C-terminal amidation (**f**). **c** Data were obtained from BioGPS and shown as Z score of the log-transformed value. **d** Data are presented as mean of frequency, and *p* values of known peptide hormones versus capped peptides is calculated by two-sided Student's *t* test. Source data are provided as a Source Data file.

cleaved from a collagen protein, and CAP-PLA2G2A (pGlu-FGEMIRLKT-NH2) which is cleaved from a phospholipase sequence (Fig. 2b).

To further understand the cellular and tissue origin of capped peptides, we next examined the mRNAs of the home genes encoding the capped peptide preproprecursors. We used BioGPS[12] as a reference

mouse tissue gene expression dataset. As shown in Fig. 2c, this set of mRNAs exhibited both cell type-specific as well as widespread tissue expression. For instance, a strong enrichment of home gene mRNAs for certain capped peptides was found in the brain (e.g., CAP-TENM1, CAP-TAC3, CAP-TAC1), in bone (e.g., CAP-EMILIN1), and in

macrophages (e.g., CAP-GDF15, CAP-PRG4). Conversely, mRNAs corresponding to other capped peptides exhibited more diffuse tissue expression across multiple cell types and organs (e.g., CAP-VIP enrichment in both brain and gut and CAP-COL5A2 expression in > 10 tissues).

Lastly, we performed more detailed amino acid composition and sequence analysis of the capped peptides from mouse plasma. As a reference and comparison set, we once again used Uniprot to manually curate a set of known mouse peptide hormones and neuropeptides (see Methods and Supplementary Data 3). Glutamine was enriched in capped peptides compared to the reference set of known peptide hormones and neuropeptides, which was expected based on our original computational search criteria. In addition, leucine was also more prevalent in capped peptides, whereas two polar amino acids (arginine, serine) were less represented (Fig. 2d). To understand whether there might be additional sequence-specific determinants of capping beyond our original N-terminal Q and C-terminal GRR/GKR motifs, we examined the amino acid sequences centered around the N- and C-termini. A modest enrichment of glycine and leucine were observed to flank both the N-terminal pyroglutamylation motif (Fig. 2e). In addition to glycine and leucine enriched around the C-terminal amidation motif, we also observed a strong enrichment for alanine at the +2 position (Fig. 2f). Together, these data demonstrate that capped peptides are produced from diverse tissues and exhibit specific patterns of amino acid composition and sequence.

## Dynamic regulation of capped plasma levels in mice

Many signaling peptides exhibit dynamic regulation in a manner dependent on internal physiologic state or external environmental conditions. We therefore measured the circulating levels of capped peptides after six distinct perturbations that spanned a wide range of physiologic processes, environmental stimuli, organ systems, and time scales: 16 h fasting vs. fed, 8-weeks high-fat diet feeding vs. chow feeding, lipopolysaccharide (LPS, 0.5 mg/kg, intraperitoneal) vs. vehicle, 6AM vs. 6PM, acute treadmill running (1 h) vs. sedentary, and 3 months vs. 24 months old. For each comparison, mouse plasma was collected and processed as described previously, and capped peptides were quantified by LC-MS (Fig. 1).

As shown in Fig. 3a, each physiologic comparison resulted in bidirectional regulation of a unique subset of capped peptides. The capped peptide/perturbation pair resulting in the most dramatic regulation was CAP-CSF1 (pGlu-LLLPKSHSWGIVLPLGELE-NH2), derived from amino acids 419-438 of full-length prepro-CSF1. Plasma CAP-CSF1 levels were induced by ~84-fold after LPS treatment ($P < 0.01$, Fig. 3a, b). Importantly, CAP-CSF1 levels were unchanged in any of the other comparisons (Fig. 3a), establishing that induction of CAP-CSF1 in plasma is a specific response to an inflammatory stimulus. Previously, the most well-known polypeptide product derived from full-length prepro-CSF1 is m-CSF1 (macrophage colony-stimulating factor 1), which is itself an LPS-inducible cytokine[13]. The co-induction of CAP-CSF1 may therefore represent additional, LPS-inducible proteolytic processing of m-CSF1. In addition, we could also identify several other interesting examples of individually regulated dynamic peptides in each of the conditions. For instance, CAP-GDNF was selectively downregulated in plasma collected at 6PM versus 6AM (pGlu-AAAASPENSRGK-NH2, 58% reduction, $P < 0.05$, Fig. 3c) and CAP-FGF5 was selectively induced by a single bout of treadmill running (1 h) versus sedentary mice (pGlu-WSPS-NH2, 2.6-fold increase, $P < 0.05$, Fig. 3d).

Beyond high magnitude changes in individual capped peptides in each condition, we also identified examples of capped peptides that exhibited coordinate regulation across multiple physiologic states. For instance, we observed a cluster capped peptides that were coordinately regulated in two distinct nutritional stressors, fasting and high-fat diet feeding. An individual example of dynamic capped peptides

within this nutrition-regulated cluster included CAP-COL27A1 (pGlu-LGPP-NH2, ~75% reduction, $P < 0.05$, Fig. 3e). The nutritional regulation of this subset of capped peptides, and of CAP-COL27A1 in particular, might point to specific functions in nutrient harvesting, fuel metabolism, or energy homeostasis. Together, we conclude that capped peptide levels in the circulation are dynamically regulated in a manner dependent on the specific capped peptide and specific physiologic perturbation.

## CAP-TAC1 is a potent agonist of mammalian tacykinin receptors

Our data so far suggest capped peptides exhibit many structural and regulatory features of other well-established peptide hormones and neuropeptides. We next sought to determine whether any of the capped peptides exhibited signaling and/or functional bioactivity. We first focused on CAP-TAC1 (pGlu-FFGLM-NH2). As we already showed in Fig. 1, CAP-TAC1 is robustly detected in blood plasma. The full-length TAC1 preprotein encodes multiple members of the tachykinin neuropeptides, including Neurokinin A/Substance K, Neuropeptide K/ Neurokinin K, Neuropeptide gamma, and Substance P (Fig. 4a)[14]. Of these known tachykinin neuropeptides, CAP-TAC1 exhibits most homology to substance P. However, a peptide with the exact chemical composition of CAP-TAC1 had not been previously reported in the literature.

We noted that the sequence of CAP-TAC1 contains the key consensus C-terminal FXGLM motif which is characteristic of all known tachykinin neuropeptides (Fig. 4a, b). In addition, the C-terminal methionyl amide, which is also present in CAP-TAC1, had previously been shown to be critical for agonist activity of other tachykinin neuropeptides[15,16]. These structural clues suggested that CAP-TAC1 might also function as a tachykinin neuropeptide-like molecule. We therefore used a cellular human TACR1-beta-arrestin recruitment assay with a fluorescence readout to directly determine the ability of CAP-TAC1 to agonize the TACR1 (also called NK1R), a high affinity receptor for substance P[17]. As shown in Fig. 4c, CAP-TAC1 exhibited dose-dependent and high potency agonism of TACR1 (EC50 = 0.7 nM). As a positive control, substance P exhibited a similar dose-dependent activation (EC50 = 1.7 nM). Both CAP-TAC1 and substance P exhibited similar levels of maximal activation (CAP-TAC1, 96.6% of maximal response; substance P, 99.8% of maximal response). A C-terminal fragment of substance P (amino acids 6-11, QFFGLM-NH2) had been previously reported to be an endogenous peptide and also an agonist at the tachykinin receptors. Substance P(6-11) differs from CAP-TAC1 in that its N-terminus is unmodified (e.g., not cyclized), but the remainder of the sequence is otherwise the same. Using the same TACR1 agonist assay, we found that Substance P(6-11) was approximately 2-fold less potent and also exhibited reduced maximal response compared to CAP-TAC1.

In addition to TACR1, there are two other mammalian tachykinin receptors, TACR2 (NK2R) and TACR3 (NK3R). Using similar cellular agonist assays for TACR2, we found that CAP-TAC1 exhibited 3-fold higher potency than the control Substance P; in addition, for this receptor, CAP-TAC1 and Substance P(6-11) were largely functionally indistinguishable (Fig. 4d). For TACR3, CAP-TAC1 was 27-fold more potent than both full-length Substance P and Substance P(6-11) (Fig. 4e). For this receptor, CAP-TAC1 also exhibited a ~20% higher maximal activation compared to Substance P(6-11). We conclude that CAP-TAC1 is a full agonist of multiple mammalian tachykinin receptors with potency similar to, or in some cases higher than, than previously established tachykinin neuropeptides. In addition, our data demonstrate that N-terminal pyroglutamylation confers functional differences in receptor engagement compared to the unmodified N-terminus.

Beyond differences in TACR activation, we reasoned that the two chemical caps of CAP-TAC1 might also produce important functional differences in terms of stability and resistance to proteolytic

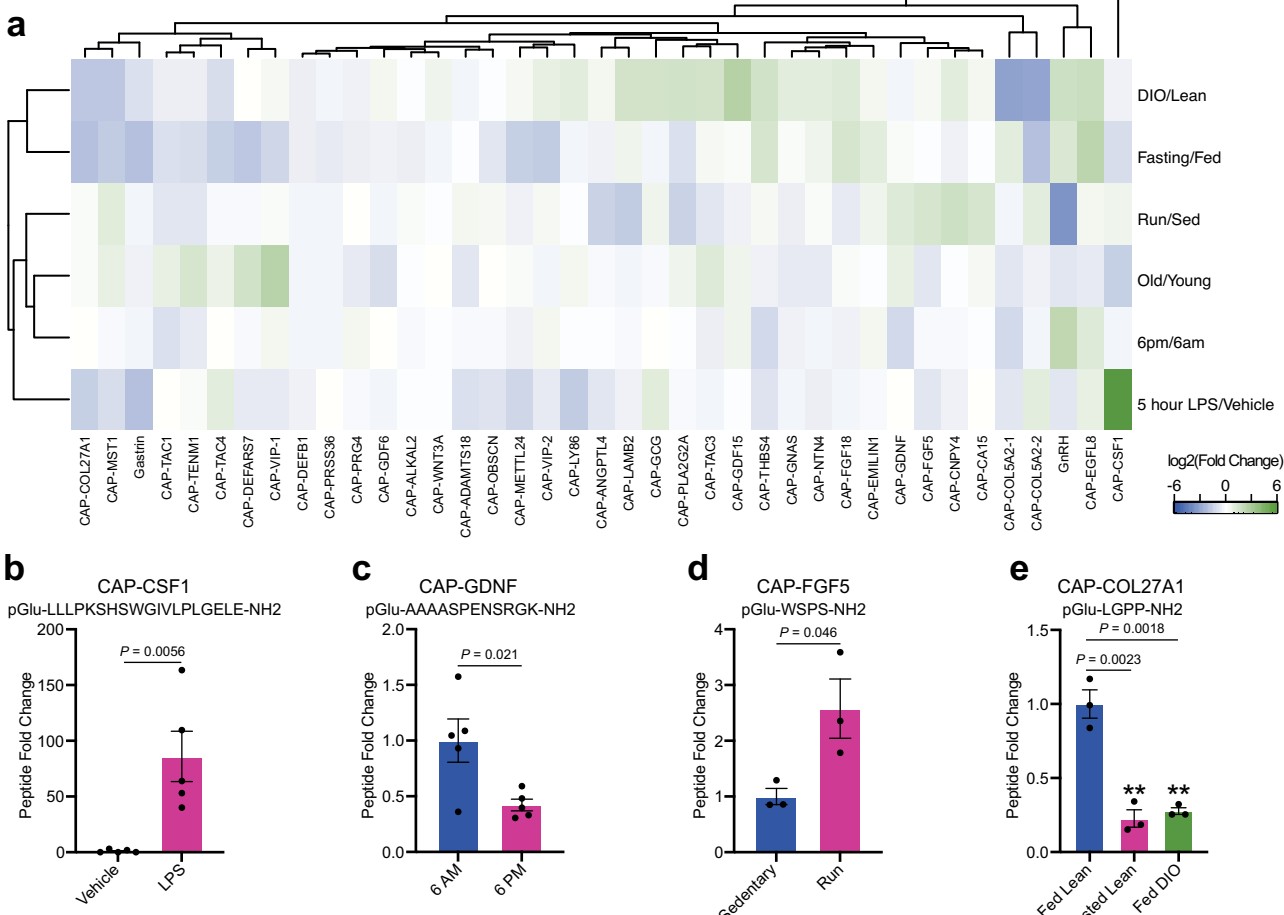

**Fig. 3 | Dynamic regulation of circulating capped peptide levels by different environmental and physiologic perturbations. a** Heatmap of the relative fold change for the indicated capped peptide in the indicated condition. **b–e** Representative peptide quantification for the indicated capped peptide in the indicated comparison. **a**, **b** "Fasting/Fed" indicates a comparison between plasma from 3-month old male mice on chow diet versus age- and sex-matched controls after fasting for 16 h ($N = 3$ biologically independent samples of pooled mouse plasma/group); "HFD/Chow" indicates a comparison between plasma from 4-month old male mice on chow diet versus 4-month old male mice after being on high-fat diet (HFD, 60% kcal from fat) for 9 weeks ($N = 3$ biologically independent samples of pooled mouse plasma/group); "LPS" indicates a comparison between plasma from 2-month old male mice following treatment with vehicle versus LPS

(0.5 mg/kg, IP, 5 h, biologically independent samples of pooled mouse plasma/group); "6AM/6PM" indicates a comparison between plasma from 2-month old male mice at the beginning of the night cycle (6AM) versus age- and sex-matched mice at the beginning of the day cycle (6PM) ($N = 5$/group); "Old/Young" indicates a comparison between plasma from 4-month old male mice versus 23-month old male mice ($N = 2$ biologically independent samples of pooled mouse plasma/group); "Run/Sed" indicates a comparison between plasma from sedentary 3-month old male mice versus age- and sex-matched controls after an acute exhaustive 60 min run on a treadmill. **b–e** Data are shown as means ± SEM. $P$ values versus control were calculated by two-sided Student's $t$ test. Source data are provided as a Source Data file.

degradation compared to Substance P. To directly test this possibility, CAP-TAC1 (10 µM) and substance P (10 µM) were individually incubated with mouse plasma and incubated at 37 °C and their levels over time were measured by LC-MS. Substance P exhibited time-dependent degradation with a $t_{1/2} = 8.5$ min. By contrast, the rate of CAP-TAC1 degradation was substantially slower ($t_{1/2} = 17.1$ min) (Fig. 4f). In fact, levels of CAP-TAC1 were still detectable after 60 min, a time point when Substance P was undetectable (Fig. 4f). Substance P(6-11) also exhibited rapid degradation kinetics which were distinct from CAP-TAC1 (Fig. 4f). Lastly, we also observed that both Substance P(6-11) and CAP-TAC1 were formed upon degradation of full-length substance P, suggesting that a combination of exopeptidase and glutamyl cyclase activity together can generate these smaller two peptides (Supplementary Fig. 4). Together, these data demonstrate that CAP-TAC1 exhibits similarities (e.g., tachykinin receptor agonism) as well as important differences (e.g., increased potency, increased plasma stability) compared to previously described tachykinin neuropeptides.

## An anorexigenic capped peptide derived from the preproprecursor region of GDF15

We next sought to understand whether signaling and bioactivity might be indeed a general feature of many capped peptides beyond CAP-TAC1 alone by performing functional studies of a second capped peptide, CAP-GDF15 (pGlu-LELRLRVAAGR-NH2, Fig. 5a). Full-length GDF15 is a secreted, 303 amino acid preproprecursor that, upon cleavage at R188, produces a C-terminal 114-amino acid anorexigenic protein hormone which is also called GDF15[18–20]. Interestingly, CAP-GDF15 mapped to amino acids 174-185, a region just upstream of the canonical GDF15 hormone and localized in the GDF15 preopropeptide region (Fig. 5a). CAP-GDF15 co-eluted with an authentic standard and a parent-to-daughter (b4+) ion transition were detected by LC-QTOF and LC-QQQ, respectively (Supplementary Fig. 1). The full MS/MS of endogenous CAP-GDF15 matched to that of a corresponding authentic standard (Supplementary Fig. 3). These data demonstrate that full-length GDF15 precursor in fact encodes at least two polypeptide products.

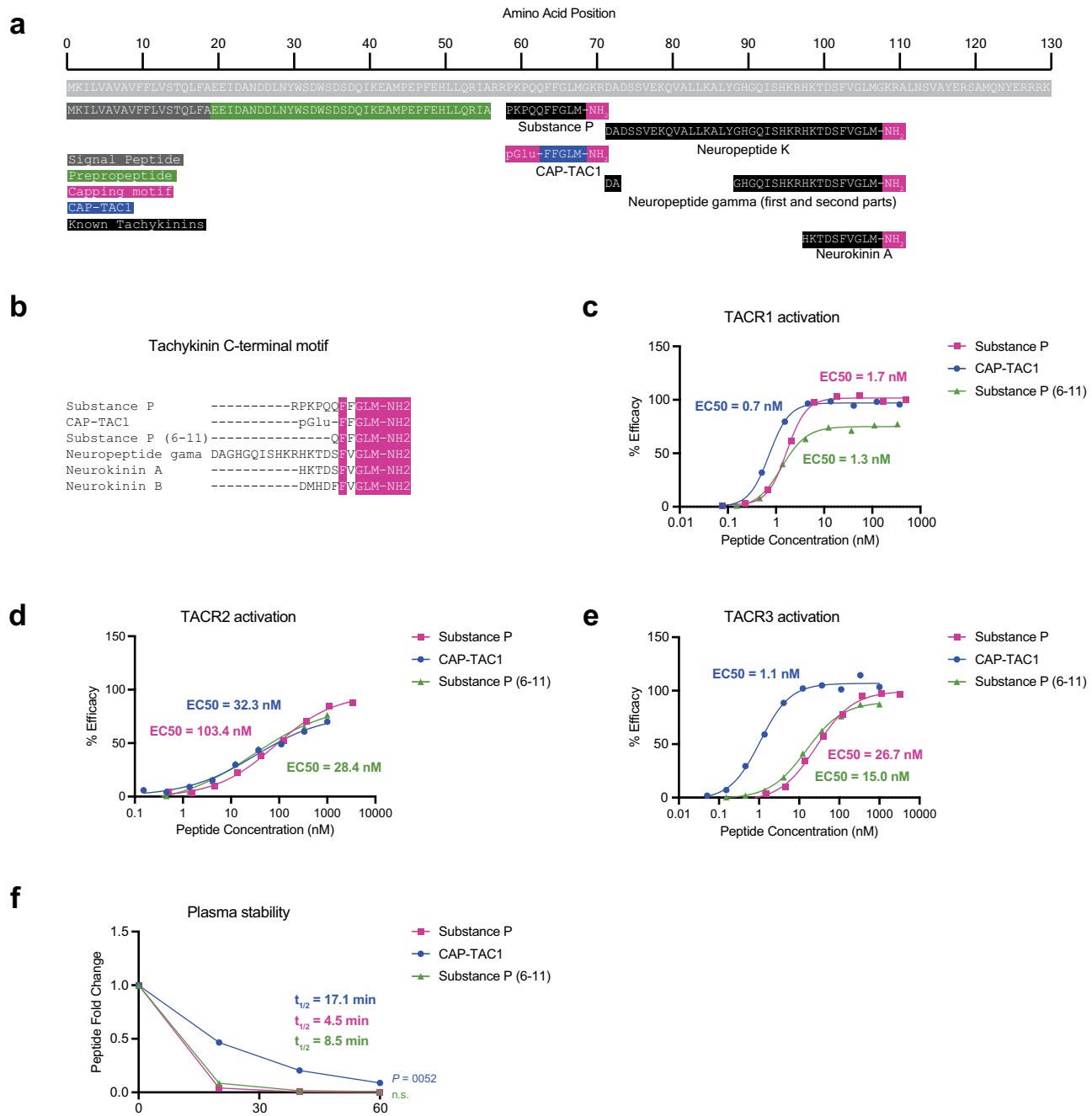

**Fig. 4 | CAP-TAC1 is a potent tachykinin receptor agonist. a** Schematic and annotation of the primary amino acid sequence for full-length mouse TAC1 pre-proprotein and its cleavage products. **b** Sequence alignment of known tachykinins and CAP-TAC1, with the C-terminal motif highlighted in pink. **c**–**e** Chemiluminescent signal intensity in PathHunter beta-arrestin CHO-K1 cells transfected with human TACR1 (**c**), TACR2 (**d**), and TACR3 (**e**) following treatment with the indicated concentration of CAP-TAC1, substance P (6-11), or substance P (SP). **f** Relative levels of CAP-TAC1, substance P (6-11), or substance P (SP) in mouse plasma following incubation at 37 °C for the indicated time. **c**–**e** $N = 2$ biological replicates per concentration and data shown as means; for (**f**), $N = 2$ biological replicates per condition and data shown as means. $P$ value were calculated with two-way ANOVA. Source data are provided as a Source Data file.

Unlike CAP-TAC1, the amino acid sequence of CAP-GDF15 did not immediately provide insights into its potential functions. However, we reasoned that the anorexigenic effects previously demonstrated by overexpression of full-length GDF15 might extend beyond the classical GDF15 hormone alone to also include CAP-GDF15. To test this possibility, we administered a single dose of CAP-GDF15 (50 mg/kg, intraperitoneally) to diet-induced obese mice and measured whole body parameters of energy balance in metabolic chambers. At this dose, concentrations of plasma CAP-GDF15 rose by 5-fold from baseline at 30 min post-administration (Supplementary Fig. 5A). CAP-GDF15 strongly suppressed food intake by ~60% compared to vehicle-treated mice (Fig. 5b). A corresponding and expected suppression of respiratory exchange ratio (RER) was also observed (Fig. 5c). Notably, CAP-GDF15 did not alter movement (Fig. 5d), oxygen consumption (VO2, Supplementary Fig. 5B), or carbon dioxide production (VCO$_2$, Supplementary Fig. 5C), demonstrating that the pharmacological effects of this peptide are specific to feeding control rather than other pathways of energy expenditure.

We next synthesized a control CAP-GDF15 peptide that preserved amino acid composition but scrambled the intervening amino acid

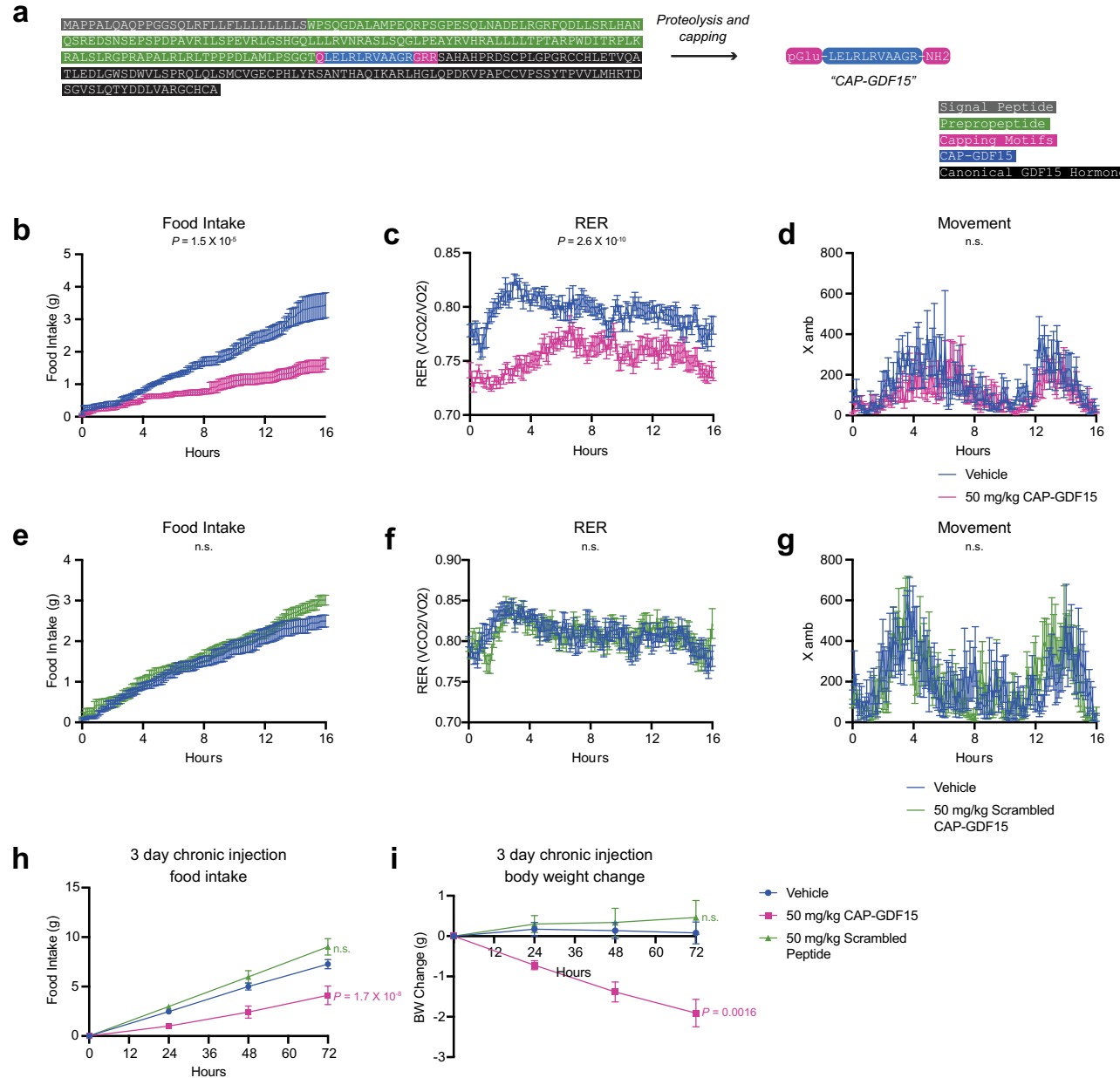

**Fig. 5 | Functional studies of CAP-TAC1 in feeding and energy balance in mice.**
**a** Schematic of the primary amino acid sequence for full-length mouse GDF15 preproprotein and annotation of CAP-GDF15 sequence as well as the canonical GDF15 hormone sequence. **b–d** Food intake (**b**), respiratory exchange ratio (RER, **c**), and ambulatory activity (**d**) of 12–16-week diet-induced obese male mice following a single treatment of CAP-GDF15 (50 mg/kg, intraperitoneal) or vehicle control. **e–g** Food intake (**e**), RER (**f**), and ambulatory activity (**g**) of 12–16-week diet-induced obese male mice following a single treatment of scrambled CAP-GDF15 (50 mg/kg, intraperitoneal, scrambled sequence: pGlu-GLEALRARLRV-NH2) or vehicle control.

**h**, **i** 3-day food intake (**h**) and body weight (**i**) of 12–16-week diet-induced obese male mice following treatment with CAP-GDF15 or scrambled CAP-GDF15 (50 mg/kg/day, IP), or vehicle control. Data are shown as means ± SEM. **b–d**, N = 12 mice/group; for **e**, **f** N = 8 mice/group; for **h** N = 7 food cages for vehicle and CAP-GDF15 groups and N = 5 for the scrambled peptide group; for **i** N = 14 mice for the vehicle group, N = 13 for the CAP-GDF15 group, and N = 5 for the scrambled peptide group. **b–i** injection occurred at time T = 0 (5:00 pm) and data were collected for the following 16 hrs. For (**b-i**), p values were calculated by two-way ANOVA. Source data are provided as a Source Data file.

sequence (scrambled CAP-GDF15, pGlu-GLEALRARLRV-NH2). This scrambled peptide control was completely ineffective in suppressing food intake and RER in metabolic chambers under identical experimental conditions (Fig. 5e–g and Supplementary Fig. 5D, E). Additionally, we synthesized an uncapped version of CAP-GDF15 (no pyro-Glu or amidation, QLELRLRVAAGR-COOH) and found this uncapped version was significantly less efficacious in reducing food intake compared to the fully capped CAP-GDF15 (Supplementary Fig. 5F). We conclude that the full anorexigenic effects of CAP-GDF15 are specific to this amino acid sequence and terminal capping modifications.

Lastly, to determine whether the acute food intake suppressive effects of CAP-GDF15 would lead to long-term suppression of feeding and obesity, we administered CAP-GDF15 or scrambled CAP-GDF15 (50 mg/kg/day, IP), or vehicle control to diet-induced obese mice. Food intake and body weight were monitored over a three-day period. A durable suppression of food intake in CAP-GDF15-treated mice was over the three-day experiment (Vehicle, 7.3 ± 0.5 g/mouse; CAP-TAC1, 4.1 ± 0.9 g/mouse/day, P < 0.05, Fig. 5h). Consequently, and as expected, an increasing reduction in body weight was also detected (Vehicle, +0.1 ± 0.3 g/mouse; CAP-GDF15, −1.9 ± 0.3 g/mouse, P < 0.001, Fig. 5i).

Importantly, mice treated with the scrambled CAP-GDF15 peptide were indistinguishable in body weight or food intake from control mice (*P* > 0.05 versus vehicle-treated mice, Fig. 5h, i). These data show that chronic CAP-GDF15 administration suppresses food intake and reduces body weight in a sequence-dependent manner. Together with CAP-TAC1, these data on CAP-GDF15 provide functional evidence for the signaling and bioactivity of two capped peptides in both cell and animal models.

### Human capped peptides and sequence comparison to mice

The capped peptide discovery pipeline described here only requires a full genome sequence and authentic peptide standards. Therefore, such an approach should also be readily amenable for discovering capped peptides in other species. Towards this end, we used the same hybrid computational-biochemical workflow as shown in Fig. 1, but now applied to protein sequences corresponding to classically secreted human proteins. Starting from $N = 3791$ secreted proteins, we predicted a total of 261 potential human capped peptides from 231 proteins (Supplementary Data 4 and 5). We synthesized authentic peptide standards by solid-phase peptide synthesis corresponding to all 261 possible human capped peptides. Once again, a two-step pipeline was performed identifying, first, the endogenous MS1 peaks in commercially available human plasma and, subsequently, a fragment ion transition with MRM methods with the same retention time as our authentic synthetic standards (see Fig. 1c, Methods, Supplementary Data 5, and Supplementary Fig. 5). In total, we provide evidence for $N = 45$ of the capped peptides with both MS1 evidence and a specific parent-to-daughter transition characteristic of the authentic standard (Fig. 6a). This number, by percentage, is similar to that previously observed with mice. In addition, we successfully acquired full MS/MS spectra for 9 human capped peptides (Fig. S7). Human capped peptides exhibited a similar distribution in plasma abundance (as quantitated using an external standard curve, Fig. 6b) and similar sequence characteristics in terms of amino acid composition (Supplementary Fig. 8A) as mouse capped peptides. Additionally, we found mass spectrometry evidence in both human and mouse plasma for the endogenous presence of 6 capped peptides with complete sequence conservation between the two species (Supplementary Fig. 8B). Using GTEx as a reference gene expression dataset, human capped peptides were also derived from preproprecursors whose mRNA levels also exhibited tissue-restricted, as well as more broad expression (Supplementary Fig. 9).

Next, we performed a multiple sequence alignment to globally understand the sequence relationship and homology across all capped peptide sequences from both mice and humans. We also performed this analysis to understand whether the human- and mouse-specific capped peptide constituted entirely distinct sequences, high homologous sequences, or some combination of these two possibilities. The resulting dendrogram is shown in Fig. 6c. We selected several subclusters as illustrative examples here. In cluster "A", we show an example of mouse and human CAP-GDNF. In this case, the sequences are largely identical but differ by only one amino acid, demonstrating the presence of species-specific, homologous sequences. The cluster labeled "B" contained three peptides, which are derived from the full-length mouse or human VIP preproprecursor. The three VIP-derived capped peptides correspond with C-terminal fragments of the known PHI-27 and VIP peptide hormones. Here, the mouse CAP-VIP-2 is identical to human CAP-VIP, and we also find evidence for an additional capped peptide which is slightly different in sequence and derived specifically from mouse prepro-VIP. These data suggest that the similar signal transduction pathways of PHI-27 and VIP peptide hormones might also extend to additional fragments of the canonical peptides. Finally, cluster "C" contains three short 3- and 5-mer capped peptides, which were amongst the shortest sequences in the entire dataset. The 3-mer capped peptides (pGlu-VL-NH2) were derived from the full-

length mouse and human FGF18 sequences and exhibited identity between the two species. The other capped peptides, derived from CNPY4 only found in mouse, constitutes a CAP-FGF18 homolog with an aspartyl-threonyl C-terminal extension (pGlu-VLDT-NH2). This cluster demonstrates that highly homolgous capped peptides can also be produced from distinct full-length preproprotein precursors. We conclude that at least a subset of the human- and mouse-specific capped peptides represent highly homologous sequences. These data also globally identify similarities as well as important differences in the sequences of capped peptides between two species.

## Discussion

Here we provide multiple lines of evidence for the endogenous presence of capped peptides, a class of previously unstudied signaling molecules. First, we provide mass spectrometry evidence that capped peptides are endogenously present in mouse and human plasma, where their levels are dynamically regulated by physiologic perturbations. Second, capped peptides exhibit post-translational N-pyroglutamyl and C-amide modifications that resemble that of other peptide hormones and neuropeptides. Third, functional studies for two capped peptides uncovered a tachykinin neuropeptide-like molecule as well as a an anorexigenic peptide, demonstrating functional bioactivity for at least two members of this class. Lastly, the majority of the precise capped peptide sequences reported here have not been previously described as chemically defined, endogenous substances in mammals. These observations suggest that N- and C-terminal "capping" defines a distinct chemical motif that is present in a large class of peptides with potential to mediate diverse axes of cell-cell communication.

Our ability to provide mass spectrometry evidence for capped peptides was enabled by a custom mass spectrometry pipeline that uses a targeted mass spectrometry approach with authentic peptide standards. This method was inspired by classical approaches in targeted small molecule metabolomics, where small molecule mass-to-charge ratios and retention times are routinely compared against synthetic standards. The generality and simplicity of this approach was demonstrated by profiling the capped peptides present in blood plasma from two different species. Importantly, such a targeted approach obviates the need for large-scale database searching. The observed concentrations of capped peptides (100 pM to 100 nM) falls within the range circulating concentration range known signaling peptides, such as gastrin, glucagon, insulin, and leptin, which are also found in blood plasma at picomolar and nanomolar concentrations. Because of potential limits of detection and potential for circulating concentrations below -100 pM, it is not unreasonable to imagine that additional capped peptides that were undetectable here might indeed be endogenously present using more sensitive mass spectrometry methods. In addition to the LC-QQQ evidence for all capped peptides, we were successful in obtaining additional complete MS/MS spectra for subset of capped peptides, including the two capped peptides that were subjected to functional validation (CAP-TAC1 and CAP-GDF15). The low abundance of these peptides remains a major experimental limitation for obtaining complete MS/MS spectra, and this is an important area for future work.

Further independent evidence strengthening the case for the endogenous presence of capped peptides is the fact that other large-scale peptidomics screens have also detected similar sequences to those reported here[21], including the VIP-derived peptide QMAVKK-LYNSILN (including the amidation)[22]. One advantage of our approach is the ability to identify very short peptides, compared to more traditional peptidomics screens which usually do not search for peptides shorter than 7–8 amino acids.

The proteolysis pathways leading to the production of capped peptides remains an important area for future work. While classical peptide hormones and neuropeptides are liberated from their

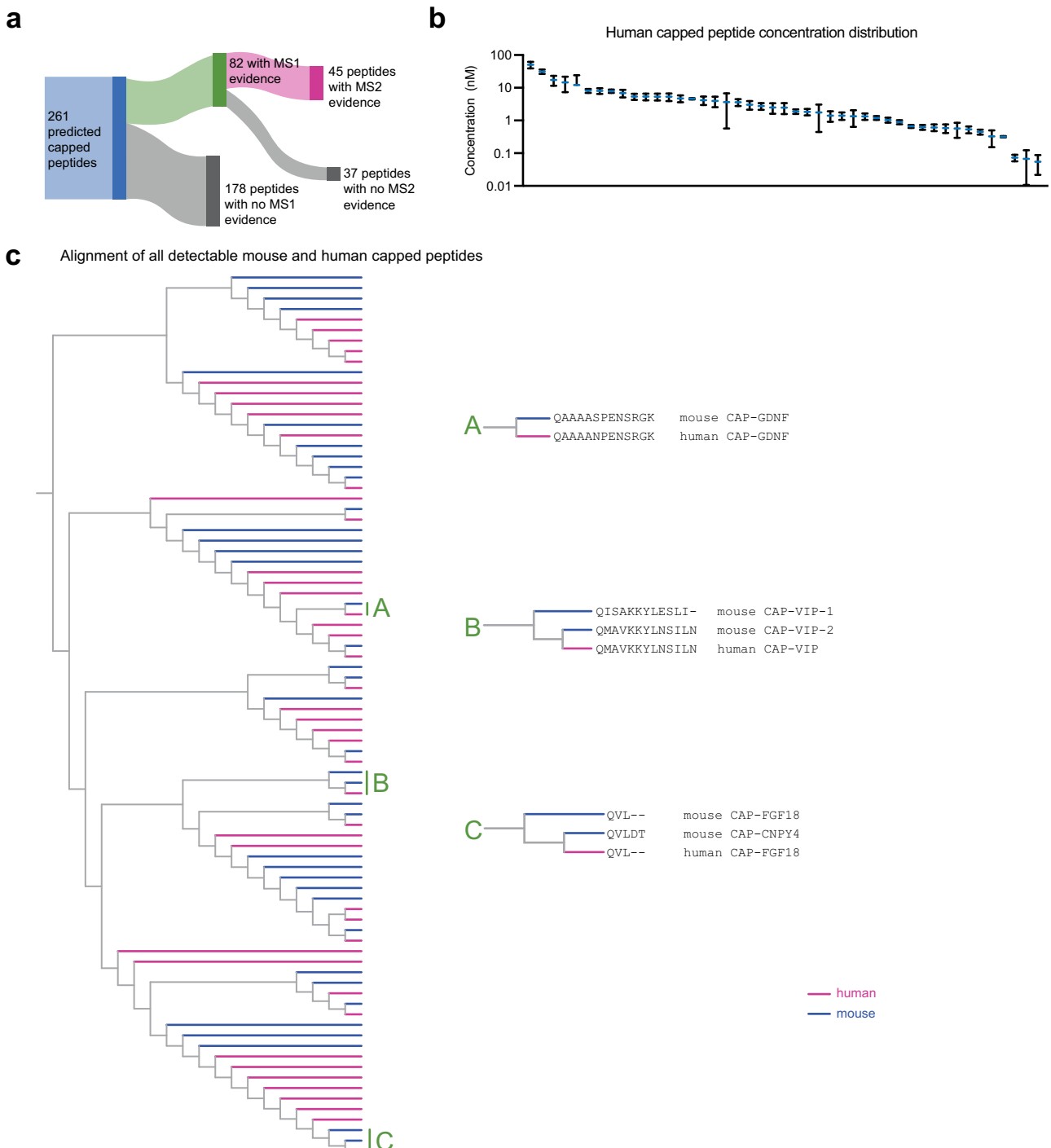

**Fig. 6 | Detection of human capped peptides and sequence alignment comparison to mouse. a** Schematic representation of proportion of capped peptides with MS1 parent ion evidence (LC-QTOF) and parent-to-daughter transition (LC-QQQ) evidence. **b** Quantification of detectable capped peptide concentrations in human plasma. **c** Phylogenetic alignment of all detectable capped peptides in human and mouse, with primary capped peptide sequences for select subclusters A–C shown on the right. For **b** data are shown as means ± SEM of *N* = 3 extractions from a vial of commercially purchased pooled human plasma. Source data are provided as a Source Data file.

preproprecursors via the action of proprotein convertases, many of the capped peptides that we detect lack an immediate upstream dibasic residue. One possibility, which we experimentally demonstrated for CAP-TAC1, is that proprotein convertase first cleave at a dibasic site further upstream of the N-terminal pyroglutamyl residue, and the resulting (longer) peptide is then trimmed via exopeptidase activity and the N-terminal pyroglutamate is subsequently installed (…**RR**PKPQ<u>QFFGLMGKR</u>…). Such a biogenesis mechanism may also

contribute to the production of CAP-GDNF (…**RR**ERNR<u>QAAAAS-</u><u>PENSRGK</u>GRR…). Others lack a proximal upstream dibasic residue; for these, we speculate that proprotein convertase-independent proteolytic mechanisms may be operational. For instance, CAP-PLA2G2A is found several amino acids C-terminal to the signal peptide of full-length PLA2G2A (signal peptide, amino acids 1–21; CAP-PLA2G2A amino acids 25-35); consequently, its N-terminus might be liberated via sequential signal peptidase and exopeptidase activity. Lastly, for

peptides including CAP-GDF15, we speculate additional proteases might be involved in liberation of the N-terminus. For instance, cathepsin L has been reported to be involved in the biogenesis of other peptide hormones/neuropeptides and exhibits very broad substrate specificity beyond basic residues alone[23,24]. Interestingly, cathepsin L and GDF15 are both highly expressed in macrophages. Moreover, proteomic studies of neo-N-termini from extracellular proteins have revealed a diversity of neo-N-terminal amino acids[25]; such non-canonical proteolysis pathways may be operational in the production of capped peptides.

Functional studies of two capped peptides, CAP-TAC1 and CAP-GDF15, provides insights into cell-cell communication in two distinct areas of signaling and physiology. Nearly all tachykinin neuropeptides discovered to date have been identified by classical biochemical purification approaches. Our identification of CAP-TAC1, and then demonstration of this molecule as a high affinity agonist of mammalian tachykinin receptors, shows that additional fragments of full-length tachykinin preproproteins may be important endogenous mediators tachykinin signaling. Like CAP-TAC1, the detection of CAP-GDF15 also demonstrates that a single full-length preproprecursor (in this case, full-length GDF15) can generate more than a single bioactive polypeptide product. Notably, CAP-GDF15 is not part of the canonical GDF15 hormone, has not been previously reported, and exhibits similar anorexigenic bioactivity to the canonical GDF15 hormone. The relative physiologic contribution of these two polypeptide products, CAP-GDF15 and canonical GDF15 hormone, from the same full-length polypeptide product remains unknown at this time. In addition, because the sequences are largely distinct, we suspect that the downstream receptor(s) of CAP-GDF15 are likely to be distinct from that of the canonical GDF15 hormone. Lastly, our initial studies of CAP-GDF15 shown here use a relatively high dose of 50 mg/kg. In the future, it will be important to establish the full dose-response and pharmacokinetic/pharmacodynamic profile of CAP-GDF15.

Beyond these two specific examples, a major future challenge and goal will be to annotate the signaling and potential functions for other capped peptides. It may be possible that a subset of capped peptides may simply be degradation fragments from other proteins, and consequently non-functional. However, our studies of CAP-TAC1 and CAP-GDF15 provide a potential roadmap for identification of those capped peptides that might exhibit bioactivity. First, several other capped peptides are similar to CAP-TAC1 in that they represent smaller fragments of known peptide hormones/signaling proteins (e.g., CAP-VIP, CAP-CSF1). It is not unreasonable to imagine that these other capped peptides might engage at the corresponding receptors and/or regulate similar physiologic processes. Second, potential functional hypotheses might arise from analysis of the physiologic functions of the full-length proteins, especially those that might not be yet explained via the action of the canonical proteins. Lastly, large-scale screening of capped peptides against a panel of candidate G-protein coupled receptors, or via functional in vitro assays may also define the fraction of bioactive capped peptides versus those that are simply inert.

## Methods

### Experimental model and subject details

**Mice and treatments.** Animal experiments were performed according to a procedure approved by the Stanford University Administrative Panel on Laboratory Animal Care. Mice were maintained in 12-h light-dark cycles at 22 °C and about 50% relative humidity and fed a standard irradiated rodent diet. Where indicated, a high-fat diet (D12492, Research Diets 60% kcal from fat) was used. Male C57BL/6 J (stock number 000664) and male C57BL/6 J DIO mice were purchased from the Jackson Laboratory (stock number 380050). For studies in high-fat diet-fed mice, peptides were dissolved in 18:1:1 (by volume) of saline:Kolliphor EL (Sigma Aldrich):DMSO and administered to mice by intraperitoneal injections at a volume of 10 μl/g at the indicated doses for the indicated times. For lipopolysaccharide injection, LPS (Sigma, #L2880-10MG) was dissolved in saline and administered to mice at a volume of 5 μl/g at indicated dose. For fasting, food was removed from mice for 16 h. For running, a six-lane Columbus Instruments animal treadmill (product 1055-SRM-D65) was used with following 1 h protocol: 10 min at 6 m/min, 50 min at 18 m/min, and increase every 2 min by 2 m/min for the last 10 minutes, all at 12° incline. For all treatment experiments, mice were mock injected with the vehicle for 3–5 days until body weights were stabilized. Heparin plasma was harvested by submandibular bleed. For all experiments, mice were randomly assigned to treatment groups. Experimenters were not blinded to groups.

**Uniprot dataset curation.** Lists of classically secreted proteins was obtained from Uniprot using the keyword "secreted" and filtering for either human or mouse species. Known peptide hormone sequences were obtained from Uniprot by first filtering for proteins annotated with keyword "hormone" for function and subsequently extracting out the specific hormone sequences from the peptides listed under the PTM annotations.

**Computational prediction of capped peptides.** Capped peptide prediction was accomplished using an in-house custom algorithm written in python (see code availability section). First, a list of classically secreted proteins was obtained from Uniprot using the keyword "secreted." Next, C-terminal amidation motifs were identified based on a GKR or GRR sequence indicative of dibasic cleavage and then amidation. N-terminal pyroglutamylation was identified by searching for Q residues within 20 amino acids upstream of the amidation motif, and capped peptides were predicted to be the inclusive sequence between the N-terminal (pyro)glutamine and the C-terminal amidation.

**BioGPS and GTExpression analysis.** For mouse, raw expression levels of precursor genes were obtained from the BioGPS dataset, GeneAtlas MOE430, gcrma https://doi.org/10.1186/1745-7580-4-5 [http://biogps.org/dataset/GSE10246/]. Replicates for the same tissue or cell type were averaged, and the relative expression was generated by normalizing the total of all tissue expression for a gene to 1. Next, the log of the relative expression was taken. An h-clustered heatmap was made with the heatmap.2 function in gplots package in R using the z-score of the log(relative expression). For human, median gene-level expression was obtained from GTEx Analysis V8 DOI: phs000424.v8.p2 [https://www.gtexportal.org/home/downloads/adult-gtex#bulk_tissue_expression] and the heatmap with h-clustered z-scores were similarly generated with heatmap.2 function in R.

**Solid-phase synthesis of capped peptides.** Capped peptides (human and mouse) were custom synthesized by Fmoc solid-phase synthesis with Rink amide resin (Elim Biopharm, Hayward, CA). The crude product was used for all synthetic standards and validated by mass spectrometry. For functional studies, peptides were purified by HPLC to >90% purity. Identity and purity were verified by MS spectra and HPLC trace (Supplemental Data 1–3).

**Plasma and authentic peptide standards preparation for peptidomics.** Plasma peptidomics preparations were adapted from Ma et al.[26]. Protease inhibitor (HALT, ThermoFisher, #78429) was added to plasma (10 μL HALT into 1 mL plasma). Plasma was diluted 1:6 plasma:Tris-HCl buffer (100 mM Tris-HCl, pH 8.2) and boiled at 95 °C for 10 minutes. In total, 1 ml of pooled plasma was used per replicate (for human plasma, Innovative Research, # IPLAWBLIH50ML). 1 mM dithiothreitol (DTT, ThermoFisher, #FERR0861) was added, samples were vortexed and incubated for 50 minutes at 60 °C. Iodoacetamide (IA, Sigma #I6125-5G) was added for a final concentration of 5 mM and incubated at room temperature for 1 hour in the dark. Formic acid

(>95%, Sigma, #F0507-500ML) was added to 0.2% final concentration. Samples were centrifuged at 21,000 × g for 20 min. Supernatants were concentrated with C8 columns (Waters, WAT054965), washed/desalted with water, and eluted in 100 µl of 80% acetonitrile. Samples were centrifuged at 21,000 × g for 10 min. Supernatant was collected for liquid chromatography-mass spectrometry (LC-MS) analysis. All authentic peptide standards were pooled into 1 ml of Tris-HCl buffer (100 mM Tris-HCl, pH 8.2) and prepared in the same way as described above.

**LC-MS detection of capped peptides.** LC-MS was performed on an Agilent 6520 Quadrupole time-of-flight LC-MS instrument. MS analysis was performed using electrospray ionization (ESI) in positive mode. The dual ESI source parameters were set as follows: the gas temperature at 325 C, the drying gas flow rate at 13 l/min, the nebulizer pressure at 30 psig, the capillary voltage at 4000 V, and the fragmentor voltage at 175 V. Separation of peptides was conducted using a C18 column (Agilent, #959961-902) with reverse phase chromatography. Mobile phases were as follows: buffer A, 100% water with 0.1% formic acid; buffer B, 90:10 acetonitrile:water with 0.1% formic acid. The LC gradient started at 95% A with a flow rate of 0.7 ml/min from 0 to 3 minutes. The gradient was then linearly increased to 40%A/60%B from 3 to 28 minutes and subsequently flushed at 5%A/95%B for 4 minutes and equilibrated back to 95%A/5%B for 6 minutes all at a flow rate of 0.7 ml/min. By LC-QTOF, a positive capped peptide detection was defined by a peak of exact mass (within 20 ppm) and co-elution (within 1 minute) of the corresponding authentic synthetic standard. The co-eluting peak was manually integrated with Agilent Mass Hunter Workstation Version 10, and concentration was estimated by standard curve integrations. Exact masses, retention times, integrations, and concentrations of all detected peptides are listed in Supplementary Data 2 and 5.

**Targeted LC-QQQ MRM and LC-IDX detection.** Targeted MRM's were obtained using Agilent 6470 Triple Quadrupole LC-MS instrument. The dual ESI source parameters were set as follows: the gas temperature at 250 C, the drying gas flow rate at 12 l/min, the nebulizer pressure at 25 psig, and the capillary voltage at 3500 V. The LC separation was done as described above. Transitions (precursor and product ions), fragmentor voltages, and collision energies for each detected capped peptide are listed in Supplementary Data 2 and 5. The MRM method was designed using MSMS spectra of the synthetic standard or the Agilent MassHunter Optimizer. Peptides were determined as detectable if they had a signal-to-noise ratio >2.5, based on previous peptidomics studies[27]. Signal-to-noise ratios were determined with Agilent MassHunter Workstation Version 10 Software. Additional full MS2 spectra were obtained for both endogenous and standard peptides with Thermo LC-ID-X with the same LC method as described above, capillary voltage at 3500 V, sheath gas at 40 Arb, auxiliary gas at 12 Arb, sweep gas at 1 Arb, ion transfer tube temperature at 325 C, vaporizer temperature at 325 C, isolation window of 1 Da, and orbitrap resolution of 50,000. The criteria for detection in a full MS/MS scan was the identification of at least one daughter ion.

**TACR agonist assay.** Dose-response curves for CAP-TAC1, Substance P (6-11), and positive control Substance P on the agonism of human TACR1/2/3 was measured by a Eurofins Discoverx using human TACR1/2/3-transfected PathHunter beta-arrestin CHO-K1 cells (Eurofins Discoverx, #493-0164).

**Half-life calculations of CAP-TAC1 in plasma.** 10 µM of either synthetic CAP-TAC1, Substance P (6-11), or SP (Sigma, #S6883) was incubated with 100 µl of mouse plasma. Samples were incubated at 37 °C for 0, 20, 40, or 60 minutes (N = 3 for each time point). At indicated time point, 1 µl of HALT protease inhibitor was immediately added, and plasma was boiled and prepared using the peptidomics workflow

described above. LC-MS spectra were obtained using Agilent 6545 Quadrupole time-of-flight LC-MS instrument as described above. Relative peptide levels were determined by total ion count area of exact mass (±50 ppm, m/z = 724.3, +1 for CAP-TAC1 and m/z = 674.4, +1 for Substance P) peak that co-eluted with synthetic standards. Basal plasma concentrations of Substance P and CAP-TAC1 were determined in samples with no synthetic peptide spiked in (N = 2) with a standard curve comparison. Half-lives were calculated using an exponential decay fit.

**Metabolic chamber studies.** For acute energy expenditure studies, food intake, RER, movement, VO2, and VCO2 were collected with CLAMS Oxymax Metabolic Cages. Mice were placed in individual metabolic cages for 24 hours prior to experiment. Mice were injected with either vehicle, 50 mg/kg CAP-GDF15, 50 mg/kg uncapped CAP-GDF15, or 50 mg/kg scrambled CAP-GDF15 at 5 pm at T = 0. Food intake, RER, movement, VO2, and VCO2 were collected every 7 minutes for 16 hours immediately after injection. All experiments were done with 3–5 month old DIO mice (Jackson, stock 380050), fed high-fat diet ad libitum.

**Sequence alignment for capped peptides.** Sequence alignment was performed with EMBL-EBI Clustal Omega tool using the guided phylotree output[28].

**Quantification and statistical analysis.** Statistical analysis was performed in Prism 9.5.1. Student's t test was used for pair-wise comparisons, and ANOVA was used for time course energy expenditure experiments were noted. Statistical significance was set at $P < 0.05$. The specific test, P value symbol and error bar meaning, definition of center, and number of replicates are noted in figure legends.

### Reporting summary

Further information on research design is available in the Nature Portfolio Reporting Summary linked to this article.

## Data availability

The LC-MS data to generated in this study to provide evidence for capped peptides in mouse and human plasma have been deposited to Mendeley Data under https://doi.org/10.17632/rcm9k9d2by.1 [https://data.mendeley.com/datasets/rcm9k9d2by/1] (raw.d mass spectrometry data from this study). The processed LC-MS integration data are available in Supplementara Data 2 and 5. All Uniprot Secretome Datasets are provided in Supplementary Data files. mRNA expression data can be obtained from BioGPS dataset, GeneAtlas MOE430, gcrma https://doi.org/10.1186/1745-7580-4-5 [http://biogps.org/dataset/GSE10246/] and GTEx GTEx Analysis V8 DOI: phs000424.v8.p2 [https://www.gtexportal.org/home/downloads/adult-gtex#bulk_tissue_expression]. All source data are provided as a Source Data files. Source data are provided with this paper.

## Code availability

Code for capped peptide prediction was deposited to GitHub https://doi.org/10.5281/zenodo.8475 [https://github.com/amandawigg/Capped-Peptides/tree/capped-peptide#capped-peptides].

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

## Acknowledgements
We thank members of the Long, Svensson, and Abu-Remaileh labs for helpful discussions. This work was supported by the NIH (DK124265 and DK130541 to J.Z.L.; DK125260, DK11916, and DK116074 to K.J.S.; GM113854 to V.L.L.), the Ono Pharma Foundation (research grant to J.Z.L.), and the Stanford Wu Tsai Human Performance Alliance (research grant to J.Z.L.).

## Author contributions
Conceptualization (A.L.W., K.J.S., J.Z.L.); methodology and software (A.L.W.); investigation A.L.W., H.Z.A., L.C., V.L.L., J.T.T., W.W., X.L.; writing—original draft (A.L.W., J.Z.L.); writing—reviewing and editing (A.L.W., H.Z.A., L.C., V.L.L., J.T.T., W.W., X.L., K.J.S, J.Z.L.), supervision (J.Z.L.), funding acquisition (V.L.L., K.J.S., J.Z.L.).

## Competing interests
A provisional patent application has been filed by Stanford University on capped peptides and use of the same. A.L.W. and J.Z.L. are listed as inventors. The remaining co-authors declare no competing interests.
