## [Peer Review File · Nature Communications]

REVIEWER COMMENTS

Reviewer #1 (Remarks to the Author):

Wiggenhorn et al. described an interesting work on a class of potential signaling molecules named "capped peptides". The peptide sequences were predicted from secreted mouse proteins and then detected in plasma by using a targeted MS approach to match the m/z and retention time with synthetic standards. Sequence and gene-level analysis of these detected peptides were performed, and the functional screening was also carried out by quantitative analysis under various physiologic perturbation. CAP-TAC1 was selected for further validation showing functions on modulation of food intake and energy metabolism. Importantly, a large number of these capped peptides were also identified in human plasma. Overall, this manuscript is significant and well-written, and most of the conclusions can be supported by the results. However, this reviewer still concerns about the identification confidence of the capped peptides as well as the biological importance of this class of biomolecules. In addition, please address the questions below.

Major issues:

1. The rationale why "capped peptides" have biological importance is not convincing. In Line 55, based on two references, the authors proposed that combinations of C- and N-terminal modifications can both increase peptide stability and enhance receptor affinity. They further explain the details in Line 99-105. However, the underlying mechanism is not clear. Why peptide stability and receptor affinity can be increased? In addition, there are many N-terminal modifications, such as N-acetylation, N-formylation, etc. Why choose N-terminal proglutamylolation?
2. It has been reported that N-terminal pyroglutamylolation could occur during sample preparation (B. Gazme et al., Food Science and Human Wellness, 2019, 8: 268-274). The N-terminal cyclization process is expedited by physical conditions such as heat and pressure. In that circumstance, the peptides are not capped in vivo. The authors need to exclude this possibility.
3. It is necessary to define the criteria of peptide matching between endogenous and the synthetic ones (Line 638-640), since false identifications could exist. For example, mass tolerance between synthetic and endogenous ones (usually less than 2 ppm)? How many fragment ions are matched (usually more than 50% b/y ions, namely 50% sequence coverage)? Sequence tags (usually more than four sequential b/y ions)? Retention time difference is less than 1 min? These criteria are essential for the identification confidence of these new capped peptides.
4. It is necessary for all the new capped peptides (64 in mouse and 85 in humans) to show mirror matching tandem mass spectra as shown in Figure 1F. The false identifications of capped peptides would lower the significance of this work as a data source.

Minor issues:

1. Figures 2F, 2G are missed labeled. It is supposed to be Figure 2E, 2F.
2. In Figure 3A, CAP-CSF1 shows the most significant change in abundance across all the physiologic perturbation, indicating a potential role in response to inflammatory stimulus. However, this peptide was not chosen for the following functional validation. Why?
3. In Figure 4E, CAP-TAC1 is missed labeled. It is supposed to be CAP-SP.
4. In Figure 5C, most of the fragment ions in high abundance are not labeled, especially m/z range larger than y7 and m/z range less than b2. If it is a tandem mass spectrum of a single peptide standard (not a chimeric spectrum), most of the fragment ions can be labeled. In addition, b7 and y7 are inconsistent in fragmentation map and spectrum labels. Therefore, the reviewer concern about the identification confidence and want to inspect the spectra of other identified peptides (64 in mouse and 85 in humans).
5. In Line 157, only C-terminal dibasic cleavage is considered in the BLASTP alignment? Why N-terminal dibasic cleavage is not considered?
6. In Line 287, "than" is duplicated.
7. In Lines 319-321, CAP-GDF15 shows function on suppressing food intake. The phenotype is significant. But, why it is not altered in Fasted/Fed of Figure 3A?

8. In Line 517 and Lines 220-224, better to perform a multiple test for Student's t test when quantifying a large number of peptides.
9. In Line 598, why use less than 20 amino acids as the peptide length cut-off? It is better to add a summary table or plot to show the lengths of the identified and predicted "capped peptides".
10. In Figure 6B, why the peptide numbers with 100% conserved sequence between mouse and human are different? Perhaps, Venn diagrams could deliver the information better than the current bar graph.

Reviewer #2 (Remarks to the Author):

Summary

In this manuscript, the authors present and investigate the hypothesis that peptides with a C-terminal amidation and an N-terminal pyroglutamation constitute a new class of orphan bloodborne peptides with bioactivity that they term "capped peptides".

They first perform an in-silico analysis in which they identify all sub-sequences in secreted mouse peptides of length 3-20 that begin with a Q and are preceded by a C-terminal GKK/GKR motif. Targeted mass-spec (comparing to synthetic standards) is then used to assay the existence and abundance of the 216 in-silico identified peptides, with detection of 30% (64 of 216 "capped peptides") in mouse blood plasma. The analysis is repeated for human sequences and blood, resulting in detection of 85 out of 260 in-silico identified "capped peptides". Many of the detected peptides across the two species are identical or nearly identical, substantiating that these peptides exist in-vivo in mammals.

Additional analyses are presented which demonstrate that the abundance of some peptides vary depending on experimental/physiological conditions such as sleep, obesity, exercise, etc. Experimental follow-up is lastly presented on selected cases. One example confirms receptor binding by a TAC1-derived peptide and another demonstrates reduction in food intake and body weight in mice administered a novel GDF15-derived peptide over 3 days.

General comments

The manuscript is well-written and the data generally seem solid. Although the main message of the manuscript is centered on the claim that a new, previously unappreciated class of capped peptides exists, to me, the most exciting scientific discovery presented is the novel GDF15-derived peptides which is demonstrated to lower food intake and body weight. This peptide seems to be truly novel in the sense that it is not part of the already known GDF15 hormone (which also lowers appetite and body weight), it has never been reported before (also not in a non-pyro-Glu variant) and comes with very exciting data demonstrating specific bioactivity, similar to the hormone GDF15 which is already known to be formed by the same protein precursor. I think they authors should give this discovery a more prominent role in the manuscript over some of the other examples presented.

Specific comments

1) In-silico analysis

It is not clear what the basis was for using GKK or GKR as the in-silico criterium for identifying C-terminally amidated peptides. There are several known bioactive peptides with C-terminal amidation formed from a GRR motif, including GLP-1, CCK8 and Orexin-B. The manuscript does not provide any argument for excluding GRR, which clearly also works with the PAM enzyme. I think the authors

should comment on or justify their choice of motif here.

What is the argument for restricting the search to peptides of length 3-20 aa when many of the well-known and characterized bioactive peptides are longer than this? The manuscript should provide some justification or explanation for this seemingly arbitrary choice of length cut-off.

2) Is the number of in-silico identified peptides surprising or not?

The category "capped peptides" is a term invented by the authors and I see no problem with that as such. But the authors make several comments in the manuscript to indicate that the number of "capped peptides" identified is in their in-silico analysis and subsequent plasma analysis are somehow greater than what should be expected or what has previously been appreciated. It would strengthen the manuscript with some additional analysis substantiating that these sequences are more prevalent than what would be expected by randomly selecting sequences from all proteins, from secreted proteins, etc. using similar criteria. For instance, are there more sequences that have a Q within 3-20 amino acids of a GKK/GKR motif, compared to a V, T, M, etc.? The authors present it as a surprise that there are 216 sequences which live up to their criteria among thousands of secreted proteins. But is that a surprise at all? What would be expected by random sampling, taking the amino acid composition of the secreted peptides into account?

3) Detection of peptides in plasma is also presented as a surprise

The manuscript also presents the detection of roughly 30% of the in-silico identified peptides in plasma as surprisingly high and as a significant discovery that substantiates the claim that this group of peptides is something special and novel. This reviewer recognizes that the data looks solid and represents a valuable scientific contribution that demonstrates the existence in blood of these peptides. But this does not necessarily mean that their existence is highly surprising or that they are all functional bioactive peptides. That aspect should be discussed more by the authors.

Firstly, many different peptidomics studies have over the past decade reported tens of thousands of distinct peptide sequences that can be detected in mammalian tissues and blood. Most of these are believed to be degradation products of mature proteins or bioactive peptides, aka not functional. Looking at the "capped" peptides that the authors highlight as novel discoveries, many of them are indeed shorter versions of peptides already known to exist. The authors should therefore discuss the possibility that many of the capped peptides detected in mouse (64) and human (85) blood could simply be degradation fragments of other peptides or proteins.

Secondly, one should expect the class of capped peptides to be more stable than other peptides in plasma (and hence enriched over randomly selected sequences) because they – by definition – harbor a C-terminal amidation which is known to stabilize peptides. So, even if these sequences were inactive fragments of bioactive peptides, they would likely still be stable enough to be detected in blood plasma. An example of this is the inactive GLP1(9-36Am) form of GLP1 which is formed when DPP4 cleaves the N-terminal end of the active GLP1(7-36Am)). Both can be detected in blood and tissues, as can shorter degraded versions. But only the full length 9-36 variant activates the receptor.

I therefore think they authors should discuss and argue more convincingly for why we should be surprised to detect 30% of the in-silico sequences in blood and be open to the possibility that many of the "capped peptides" may be non-functional degradation products.

Lastly, I would suggest a comparison of the sequences identified in plasma by the authors with peptides detected in other large peptidomics screens in tissues and blood, to show the reader if these peptides are novel discoveries in the sense that they have never been reported before (with and without the pyro-Glu).

4) Novelty of the CAP-TAC1 case story

To my surprise, the CAP-TAC1 findings are highlighted as the main case-story in the manuscript where it is presented as a novel peptide which binds to TACR1/NK1R. For instance, one sentence in the discussion reads "Interestingly, CAP-TAC1 is only 5 amino acids in length and therefore represents the shortest naturally-occurring tachykinin receptor agonist reported to date in any species."

This reviewer finds the wording a bit inflated and the novelty a bit overstated, considering that many other shorter versions are known to be bioactive, including a sequence identical to CAP-TAC1, just without the pyro-Glu.

Here, I think the authors should present the data in a way that makes it easier for the reader to assess the novelty and impact of their discoveries in an area where there is already a lot of prior work demonstrating activity of shorter variants of SubstanceP.

The novel CAP-TAC1 sequence is pGlu-FFGLM-Am which corresponds exactly to the C-terminal end of Substance P. It also has very high similarity to the C-terminal ends of Neurokinin A and B.

Substance P (TAC1 derived): RPKPQQFFGLM-Am
Neurokinin A (TAC1 derived): HKTDSFVGLM-Am
Neurokinin B (TAC3 derived): DMHDFVGLM-Am

The fact that this shorter sequence identified by the authors (CAP-TAC1) has affinity for the same receptors as SubstanceP is not that surprising to this reviewer. Especially, considering that previous work (cited by the authors) has shown that the non-pyro version of their "novel" CAP-TAC1 peptide (QFFGLM-Am) is a potent agonist of the tachykinin receptors and that other highly similar sequences such as FGLM-Am and pGlu-FFPLM-Am also work.

Given this data, the authors should provide a figure showing the many highly similar sequences already known to be bioactive and demonstrate that their pyro-Glu variant is different in terms of activity to the non-pyro-Glu variant which has already been shown to be active too.

5) Are "capped peptides" a novel class of peptides?

The two modifications (N-term PyroGlu and C-term amidation) are both well known to 1) be formed in vivo (enzymes exist for creating both) 2) to be important for stability and function. The novel idea here should therefore be that the combination constitutes a special class of endogenous peptides that has hitherto been overlooked.

I think the authors should argue a bit better for this point, or alternatively revise the wording in their manuscript. Many of the sequences seem to just be shorter versions of already known peptides with a C-terminal amidation. What is lacking is a demonstration of two things:

A) the uniqueness of this particular shorter variant. It would strengthen the manuscript with a few examples where the authors demonstrate experimentally that slightly longer or slightly shorter versions of these peptides do not exist in vivo, aka that only the variant which start with a Q is present in blood.

A lot of previous peptidomics papers have found "ladders of degradation" where pretty much all shorter fragments exist in-vivo, pointing to a degradation mechanism where, for instance, the N-terminal is gradually degraded. Since the authors only look for one variant with their targeted approach, they are "blind" to the potential existence of other fragments that may be equally present and/or active.

B) that the pyro-Glu variant is indeed more stable and more bioactive than the "raw" Q variant, and more stable/active than slightly longer or shorter sequences. Unless "capped peptides" stand out distinctly from other related variants in terms of abundance and bioactivity then there is no strong basis for claiming that these represent a special, biologically relevant sub-category of peptides that others in the field should be interested in.

As the authors point out themselves, at least 3 capped peptides are already well characterized, so the conceptual idea that a double-capped peptide with pyro-Glu and Amidation exist is not novel. The novelty claimed here is that they are more abundant, more important, etc. than previously recognized, and this reviewer does not think that the manuscripts argues sufficiently strong for that.

6) The method works for finding bioactive peptides...or at least for finding one exciting example

Having said all of the above, the method does identify the GDF15-derived peptides which seems to be novel both in terms of sequence, detection and function. The fact that this peptide is NOT a sub-fragment of the GDF15 hormone itself but rather a C-terminal fragment of what is currently believed to be a pro-peptide of the same precursor make this a very interesting finding. In that sense, the GDF15 story is much more novel and surprising than the CAP-TAC1 story. The fact that this new peptide reduces food intake and lowers body weight in-vivo makes it a discovery worthy of publication in itself. It would suggest to give this story a more prominent place in the manuscript, since this – to this reviewer – represents the most significant scientific discovery in this work.

7) Formation of pyro-Glu

It is generally assumed in the manuscript that because the pyro-Glu peptides are detected in blood, they also exist in-vivo.

This is not an unreasonable assumption given that (in humans) the pyro-Glu conversion is catalyzed by two isoforms of Glutaminyl cyclases (<https://www.sciencedirect.com/science/article/pii/S0014299922004393>), one secreted and one located in the Golgi.

Unless the conversion is spontaneous, these enzyme must come into proximity of the proposed capped peptides, which could occur both in the secretory pathway or extracellularly.

However, spontaneous pyro-Glu formation has been described and discussed in the literature and has formation as an experimental artifact. How sure are the authors that the detected pyro-Glu capped peptides are in fact present in-vivo and not an artifact of the experimental analysis? There are reports which seem to indicate that these modification can occur as experimental artifacts:

<https://pubs.acs.org/doi/10.1021/ac501451v> and

<https://pubs.acs.org/doi/10.1021/acsomega.9b04384>. It would strengthen the manuscript with some additional arguments for the in-vivo relevance and/or more discussion of the possibility of these findings being artifacts.

In the context of this discussion, it would again be relevant for the authors to demonstrate that the pyro-Glu variant is indeed more stable or more active than its non-modified counterpart and/or that slightly longer or shorter variants are not stable and active too. That would strengthen the case for the functional importance of the pyro-Glu modification and rule out the possibility that the stability and function is mainly or solely driven by the amidation.

Reviewer #3 (Remarks to the Author):

The manuscript describes the use of bioinformatics to predict a specific class of peptide that has a C-

terminal amide and an N-terminal pyroglutamate. The authors then synthesised a large number of the theoretical peptides and then assessed if they were present in the circulatory peptidome of mice. Whilst this is a possible avenue for detecting and identifying bioactive peptides, I am concerned that the peptides they have selected for further study are not likely to be produced endogenously. I have highlighted some areas of concern I have with the manuscript.

Peptide selection, and potential route of endogenous production.

Most bioactive peptides in mammalian systems have basic amino acids present in or near their theoretical cleavage sites. Whilst the reviewers do discuss this, the main two peptides they discuss are cleaved from their prohormone sequences without these basic amino acids immediately upstream of the N-terminal – or in the case of CAP-TAC1, there are basic amino acids two residues upstream from the proposed cleavage site.

The CAP-TAC1 peptide from the mouse described in the paper – shown below seems to have two amino acids upstream on the N-terminus in front of the proposed n-terminal pyroglutamate of the peptide (pQFFGLM-NH₂). This peptide could still possibly be present, however would need to be produced after the original PQFFGLM peptide was produced and subsequently cleaved by an enzyme such as DPP-IV. Or substance P itself was produced and had four amino acids removed enzymatically. The same sequence is obviously present in human – with the additional PQ on the N-terminus of the proposed peptide sequence.

Mouse TAC1 sequence:

MKILVALAVFFLVSTQLFAEEIGANDDLNYSDWYDSDQIKEELPEPFEHLLQRIARRPKP**QFFGLM**GKRDADSSI
EKQVALLKALYGHGQISHKRHKTD SFVGLMGKRALNSVAYERSAMQNYERRR

Human sequence:

MKILVALAVFFLVSTQLFAEEIGANDDLNYSDWYDSDQIKEELPEPFEHLLQRIARRPKP**QFFGLM**GKRDADSSI
EKQVALLKALYGHGQISHKRHKTD SFVGLMGKRALNSVAYERSAMQNYERRR

The same issue is apparent for the GDF15 sequences that are described in the manuscript. The mouse GDF15 sequence for CAP-GDF15 doesn't have any sequence for the peptide to be cleaved from the prohormone sequence upstream of the initial N-terminal pyroglutamate of CAP-GDF15, and the human sequence has a RP sequence immediately upstream, which is unlikely to result in a preferred cleavage site by standard prohormone convertases. Ghrelin has a PR sequence at the C-terminus, which results in its cleavage from the prohormone sequence, but a RP is unlikely to be cleaved.

Human GDF-15 sequence

MPGQELRTVNGSQMLLVLLVLSWLPHGALS LAEASRASFPGPSELHSEDSRFRELKRKYEDLLTRLRANQSWEDS
NTDLVPAPAVRILTPEVRLGSGHHLRISR AALPEGLPEASRLHRALFRLSPTASRSWDVTRPLRRQLSLARPQAPAL
HLRLSPPPSQSDQLLAESSARP**QLELHLRPAAR**GRRRARARNGDHCPGPGRCCLHTVRASLEDLGWADWV
LSPREVQVTCIGACPSQFRAANMHAQIKTSLHRLKPDTPAPCCVPASYNPMVLIQKTD TGVS LQTYDDLAKDCH
CI

Mouse GDF-15 sequence

MAPPALQAQPPGGSQRLRFLFLLLLLLLLSWPSQGDALAMPEQRPSGPESQLNADELGRFRQDLSRLHANQSREDS
NSEPSPDPAVRILSPEVRLGSHGQLLRVNRASLSQGLPEAYRVHRALLLPTARPWDITRPLKRALSRLGPRAPALR
LRLTPPDLAMLPSGGT**QLELRLRVAAGR**GRRSAHAHPRDSCPLGPGRCCHLETQATLEDLGWSDWVLSRQLQ
LSMVCVGECPHLYRSANTHAQIKARLHGLQPKVPAPCCVPSSYTPVVLHRTDSGVSLQTYDDLVAR GCHCA

As can be seen from the above mouse GDF15 sequence, there are no basic amino acids upstream of the proposed peptide sequence, therefore the peptide is unlikely to be formed as part of the normal bioactive peptide cleavage process. The authors need to discuss the proposed cleavage mechanism for this peptide from the GDF15 precursor sequence. The authors should look at the cleavage positions of the other peptides they discuss (such as GDNF, FGF5, PLA2G2A) and discuss how these peptides would be produced endogenously.

With regards to the supplied evidence of confirmation of detected endogenously produced peptides, I feel that this needs to be strengthened significantly.

pQFFGLM-NH₂ peptide supplied data.

The supplied images in figure 1 (panel E and F) and S1 do not look like raw data exported from an MS software, however looks like the data has been reproduced in a different program. This should be re-done to include original data. The spectra in figure 1F only shows 1 decimal place – and obviously this value has been rounded. Current mass spectrometers – including the one used by the authors – are capable of much higher accuracy. The MS/MS spectra of the peptides detected in an in-vivo setting and their in-vitro generated equivalents should be reproduced in much clearer fashion, so that the readers can definitely see that the peptides match.

pQLELRVAAGR-NH₂ peptide supplied data.

The supplied images in figure 5 (panel B and C) again does not look like raw data exported from an MS software. This should be re-done to include original data. Again - the spectra only shows 1 decimal place. It seems that there only MS/MS spectral data is displayed up to the precursor m/z value, was data not acquired above this the precursor m/z value? – this would give more data for comparing the endogenous and synthetic peptides.

If the authors are claiming to detect these peptides endogenously, more data should be provided – such as demonstrating that the precursor ions match exactly, proper chromatographic comparison and product ion spectral matching. This data should not be retouched using a separate program, but should come directly from the MS software packages used to look at the data.

Issues surrounding the dosage of the GDF15 derived peptide

Dosage of the GDF15 peptide to animals was performed at 50mg/kg – as long as this isn't a typographic error and should be 50µg/mL, this level of dosing is absolutely immense for a potentially bioactive peptide. Most bioactive peptides are dosed at far lower amounts in pre-clinical studies – usually in the 1-3 mg/kg range. The plasma levels of these dosed peptides would be in the very high µg/mL, which is significantly above what levels of the endogenous peptides would reach. GDF15 is present in mouse plasma at approximately 100 pg/mL – a quick google found this article <https://doi.org/10.1038/s41598-019-56922-w> the authors should indicate what the concentration of their CAP-GDF15 peptide reached in plasma and why it was dosed at such a high level. The authors need to discuss the reason for the large dose and highlight that this level would be significantly above any potential endogenous level.

To confirm that these identified peptides are indeed produced endogenously, then the authors need to compare the concentration of the peptide they are identifying and compare these to the expected concentration in the animal. For example, endogenous levels of substance P is in the low tens of picograms per millilitre – what is the concentration of the level of the capped peptide that was detected? If these concentrations are comparable with the expected concentration of the more classical prohormone derived protein, then it would add weight to their potential as bioactive peptides.

Minor issues

The sensitivity that the authors have achieved for the peptide analysis is very good for a high flow analyses (0.7 mL/minute). Looking at the extraction methodology, it is unclear how much plasma was extracted – was it 100 µL or 1mL – please clarify.

REVIEWER COMMENTS

Reviewer #1 (Remarks to the Author):

Wiggenhorn et al. described an interesting work on a class of potential signaling molecules named “capped peptides”. The peptide sequences were predicted from secreted mouse proteins and then detected in plasma by using a targeted MS approach to match the m/z and retention time with synthetic standards. Sequence and gene-level analysis of these detected peptides were performed, and the functional screening was also carried out by quantitative analysis under various physiologic perturbation. CAP-TAC1 was selected for further validation showing functions on modulation of food intake and energy metabolism. Importantly, a large number of these capped peptides were also identified in human plasma. Overall, this manuscript is significant and well-written, and most of the conclusions can be supported by the results. However, this reviewer still concerns about the identification confidence of the capped peptides as well as the biological importance of this class of biomolecules. In addition, please address the questions below.

Major issues:

1. The rationale why “capped peptides” have biological importance is not convincing. In Line 55, based on two references, the authors proposed that combinations of C- and N-terminal modifications can both increase peptide stability and enhance receptor affinity. They further explain the details in Line 99-105. However, the underlying mechanism is not clear. Why peptide stability and receptor affinity can be increased? In addition, there are many N- terminal modifications, such as N-acetylation, N-formylation, etc. Why choose N-terminal proglutamylation?

We agree that further clarification should be provided about the rationale for investigating “capped peptides” as signaling molecules. We have therefore added the following **new text** to the Introduction:

“From a chemical perspective, a subset of mammalian neuropeptides/peptide hormones are usual in that they contain co-incident N-terminal pyroglutamyl and C-terminal amide post-translational modifications. Representative examples include TRH (pGlu-HP-NH₂) and GnRH (pGlu-HWSYGLRPG-NH₂). [...] Beyond this subset of neuropeptides and peptide hormones, co-incident N-pyroglutamyl/C-amide modifications have not been identified, suggesting that they appear to be restricted to and designate a subset of privileged sequences that encode for bioactive signaling peptides. [...] We hypothesized that such peculiar and co-incident N-pyroglutamyl/C-amidation modifications of peptides are not installed by happenstance, but instead defines a chemical motif that designates a much larger set of potentially bioactive signaling peptides than has been reported to date. This notion was inspired by the well-established observation that certain chemical motifs are already define classes of molecules and functions. For instance, a free amino group is characteristic of monoamines; a cyclized arachidonate acid is characteristic of prostanoids, and a cholesterol backbone is characteristic of steroids.”

We also agree that additional clarification should be provided about the current knowledge of how capping increases peptide stability and receptor affinity. We have added the following **new text** to the Introduction: “For instance, removal of both terminal modifications of TRH renders the resulting unmodified peptide devoid of agonist activity at the TRH receptor and highly sensitive to proteolytic degradation.⁷”

Lastly, we agree with the referee's suggestion that other N-terminal modifications could be potentially explored. To discuss this possibility, and our rationale for initially focusing on pyroglutamate, we have added the following **new text** to the Discussion: "Projecting forward, considering that many other modifications of peptide N-termini have been reported, including N-acetylation and N-formylation, we suspect that the methodology presented here may be more generalizable beyond the set of capped peptides presented here. [...] Exploration of the generality of this methodology remains an important area for future work."

2. It has been reported that N-terminal pyroglutamylation could occur during sample preparation (B. Gazme et al., Food Science and Human Wellness, 2019, 8: 268–274). The N-terminal cyclization process is expedited by physical conditions such as heat and pressure. In that circumstance, the peptides are not capped in vivo. The authors need to exclude this possibility.

We thank the reviewer for this valuable suggestion. We have now generated **new data** to exclude the possibility of artefactual formation of pyroglutamate, which are presented in a **new Fig. S2A**. We have added the following **new text** to describe these data: "To exclude the possibility that N-pyroglutamylation may be artefactually occurring in the sample preparation, we subjected a synthetic standard of uncapped CAP-TAC1 (QFFGLM) to the sample preparation conditions. As shown in **Fig. S2A**, we did not observe any formation of pGlu-FFGLM from the QFFGLM starting material."

N- or C-cap formation with peptidomics
preparation of uncapped CAP-TAC1

3. It is necessary to define the criteria of peptide matching between endogenous and the synthetic ones (Line 638-640), since false identifications could exist. For example, mass tolerance between synthetic and endogenous ones (usually less than 2 ppm)? How many fragment ions are matched (usually more than 50% b/y ions, namely 50% sequence coverage)? Sequence tags (usually more than four sequential b/y ions)? Retention time difference is less than 1 min? These criteria are essential for the identification confidence of these new capped peptides.

We agree that clearer definitions should be presented for the criteria for peptide matching. Originally, our criteria for LC-QTOF with synthetic standards was by retention time (within 1 min) and m/z (within 20 ppm). The retention time window was used to account for inter-sample retention time drift; the m/z window was set at 20 ppm because we typically observed 2-14 ppm mass accuracy with our synthetic standards using our instrument. We have added the following **new text** to the Methods to describe these criteria: "By LC-QTOF, a positive capped peptide detection was defined by a peak of exact mass (within 20 ppm) and co-elution (within 1 minute) of the corresponding authentic synthetic standard."

In additional **new experiments**, we have performed additional targeted measurements using multiple-reaction-monitoring on an LC-QQQ instrument. Notably, these methods enable (at unit mass) isolation, fragmentation, and measurement of specific parent-to-daughter transitions and provide important independent validation of our original LC-QTOF data. Here, our positive detection criteria were defined by retention time (within 1 min) and a transition with signal-to-noise ratio > 2.5. This is similar to what has been used in other papers in the field of targeted peptidomics (e.g., Donohue *et al.*, 2021). We have added the following **new text** to the Methods to describe these criteria: “**Targeted LC-QQQ MRM Detection.** Targeted MRM’s were obtained using Agilent 6470 Triple Quadrupole LC-MS instrument. The dual ESI source parameters were set as follows: the gas temperature at 250C, the drying gas flow rate at 12 l/min, the nebulizer pressure at 25 psig, and the capillary voltage at 3,500 V. The LC separation was done as described above. Transitions (precursor and product ions), fragmentor voltages, and collision energies for each detected capped peptide are listed in **Tables S2 and S5**. The MRM method was designed using MSMS spectra of the synthetic standard or the Agilent MassHunter Optimizer. Peptides were determined as detectable if they had a signal-to-noise ratio greater than 2.5, based on previous peptidomics studies (Donohue *et al.*, 2021). Signal-to-noise ratios were determined with Agilent MassHunter Software.”

Lastly, we have added a **new figure** to show raw chromatograms of LC-QTOF and LC-QQQ detection for each capped peptide in mouse (**Fig. S1**) and human (**Fig. S5**). A representative subset of these is included here, and we refer the reviewer to those Supplementary Figures for the remaining panels.

For clarity, and to help focus the detection issue on the targeted MRM methods, we have now removed the mirror plots originally presented in Fig. 1 and 5.

4. It is necessary for all the new capped peptides (64 in mouse and 85 in humans) to show mirror

matching tandem mass spectra as shown in Figure 1F. The false identifications of capped peptides would lower the significance of this work as a data source.

As mentioned in response to this reviewer's question #3, our evidence for positive detection relies on targeted MRM transitions that are characteristic of the authentic peptide standards. We have elected to use the LC-QQQ approach as secondary confirmation (rather than showing all mirror plots) because of the significantly higher sensitivity of triple quadrupole analysis. We have therefore provided all transitions and collision energies in the **Table S2** and **Table S5**; in addition, we have generated a **new figure** showing raw chromatograms of the LC-QTOF and LC-QQQ methods for all detected capped peptides (**Fig. S1** and **Fig. S5**).

Minor issues:

1. Figures 2F, 2G are missed labeled. It is supposed to be Figure 2E, 2F.

Thank you, we have modified the labeling accordingly.

2. In Figure 3A, CAP-CSF1 shows the most significant change in abundance across all the physiologic perturbation, indicating a potential role in response to inflammatory stimulus. However, this peptide was not chosen for the following functional validation. Why?

We thank the reviewer for this suggestion. We are actively exploring the role of CAP-CSF1 in immune system function, but this is a larger body of ongoing work which currently remains preliminary and incomplete. For now, we have added the following sentences to the discussion: "Several other capped peptides are similar to CAP-TAC1 in that they represent smaller fragments of known peptide hormones/signaling proteins (e.g., CAP-VIP, CAP-CSF1). It is not unreasonable to imagine that these other capped peptides might engage at the corresponding receptors and/or regulate similar physiologic processes."

3. In Figure 4E, CAP-TAC1 is missed labeled. It is supposed to be CAP-SP.

Thank you. We have fixed the labeling in Figure 4.

4. In Figure 5C, most of the fragment ions in high abundance are not labeled, especially m/z range larger than y7 and m/z range less than b2. If it is a tandem mass spectrum of a single peptide standard (not a chimeric spectrum), most of the fragment ions can be labeled. In addition, b7 and y7 are inconsistent in fragmentation map and spectrum labels. Therefore, the reviewer concern about the identification confidence and want to inspect the spectra of other identified peptides (64 in mouse and 85 in humans).

As stated in response to this reviewer's question #3, we have now removed the mirror plots originally presented in Figs. 1 and 5. We have instead added new targeted MRM detection of capped peptides, which are shown in **Fig. S1** and **Fig. S5**.

5. In Line 157, only C-terminal dibasic cleavage is considered in the BLASTP alignment? Why N-terminal dibasic cleavage is not considered?

We did not use an N-terminal dibasic residue as a search criterion because of the precedent of other N-pyroglutamyl/C-amide peptides. For instance, an "SS" sequence is found upstream of the N-pyroglutamyl in mGnrh1 (N-LEGCS**SS**QHWSYGLRPGGK-C). To clarify this point, we have added the following **new text** to the Results: "Our search criteria were designed to use genome

sequence information to “re-discover” known N-pyroglutamyl/C-amide peptides, and also uncover any additional potential peptides that, by sequence, might also contain similar chemical capping motifs.”

6. In Line 287, “than” is duplicated.

The duplicate word has now been deleted.

7. In Lines 319-321, CAP-GDF15 shows function on suppressing food intake. The phenotype is significant. But, why it is not altered in Fasted/Fed of Figure 3A?

We interpret these data to demonstrate that the physiologic regulation of the levels of capped peptides may or may not be correlated with their functional effects. It is not unreasonable to imagine that certain capped peptide with physiologic regulation may (or may not) be functional, and conversely, those that exhibit robust functional bioactivity may (or may not) be dynamically regulated.

8. In Line 517 and Lines 220-224, better to perform a multiple test for Student’s t test when quantifying a large number of peptides.

To clarify, in these initial survey analyses and the heat map shown in **Fig. 3A** we generated using only fold change as the criteria to prioritize the most dynamically regulated capped peptides from the entire dataset. The Student’s t-test is used in the small number of individual comparisons shown in **Fig. 3B-E**. For clarity, we have removed the sentences in the lines indicated by the reviewer (“Across all measurements... 2-fold.”).

9. In Line 598, why use less than 20 amino acids as the peptide length cut-off? It is better to add a summary table or plot to show the lengths of the identified and predicted “capped peptides”.

The length criteria used in our computational method was selected so that known N-pyroglutamyl/C-amide peptides, including TRH (3-mer), GnRH (10-mer), and gastrin (17-mer) could be “re-discovered.” As mentioned above, we have added the following new text in the Results to discuss this: “Our search criteria were designed to use genome sequence information to “re-discover” known N-pyroglutamyl/C-amide peptides, and also uncover any additional potential peptides that, by sequence, might also contain similar chemical capping motifs.”

Table S2 now contains the requested information for the table showing the lengths of predicted and detected capped peptides.

10. In Figure 6B, why the peptide numbers with 100% conserved sequence between mouse and human are different? Perhaps, Venn diagrams could deliver the information better than the current bar graph.

Thank you for this suggestion. For clarity, we have removed **Fig. 6B-D**, and replaced it with a **new figure Fig. S6A**. Now, we present a Venn diagram of the peptides that are detected in both human and mouse plasma. We have edited the text as well: “Additionally, 6 capped peptides, complete sequence conservation between humans and mice, were detected in both human and mouse plasma (**Fig. S6A**).”

Reviewer #2 (Remarks to the Author):

Summary

In this manuscript, the authors present and investigate the hypothesis that peptides with a C-terminal amidation and an N-terminal pyroglutamation constitute a new class of orphan bloodborne peptides with bioactivity that they term “capped peptides”.

They first perform an in-silico analysis in which they identify all sub-sequences in secreted mouse peptides of length 3-20 that begin with a Q and are preceded by a C-terminal GKK/GKR motif. Targeted mass-spec (comparing to synthetic standards) is then used to assay the existence and abundance of the 216 in-silico identified peptides, with detection of 30% (64 of 216 “capped peptides”) in mouse blood plasma. The analysis is repeated for human sequences and blood, resulting in detection of 85 out of 260 in-silico identified “capped peptides”. Many of the detected peptides across the two species are identical or nearly identical, substantiating that these peptides exist in-vivo in mammals. Additional analyses are presented which demonstrate that the abundance of some peptides vary depending on experimental/physiological conditions such as sleep, obesity, exercise, etc. Experimental follow-up is lastly presented on selected cases. One example confirms receptor binding by a TAC1-derived peptide and another demonstrates reduction in food intake and body weight in mice administered a novel GDF15-derived peptide over 3 days.

General comments

The manuscript is well-written and the data generally seem solid. Although the main message of the manuscript is centered on the claim that a new, previously unappreciated class of capped peptides exists, to me, the most exciting scientific discovery presented is the novel GDF15-derived peptides which is demonstrated to lower food intake and body weight. This peptide seems to be truly novel in the sense that it is not part of the already known GDF15 hormone (which also lowers appetite and body weight), it has never been reported before (also not in a non-pyro-Glu variant) and comes with very exciting data demonstrating specific bioactivity, similar to the hormone GDF15 which is already known to be formed by the same protein precursor. I think they authors should give this discovery a more prominent role in the manuscript over some of the other examples presented.

We thank the reviewer for this suggestion and completely agree that CAP-GDF15 should be emphasized. We have therefore made the following text modifications in the Abstract: “A second capped peptide, CAP-GDF15, is a previously unknown 12-mer peptide cleaved from the prepropeptide region of full-length GDF15 that, like the canonical GDF15 hormone, also reduces food intake and body weight.”

To underscore the studies of CAP-GDF15, we have added the following text to the Discussion: “Like CAP-TAC1, the detection of CAP-GDF15 also demonstrates that a single full-length preproprecursor (in this case, full-length GDF15) can generate more than a single bioactive polypeptide product. Notably, CAP-GDF15 is novel in the sense that it is not part of the canonical GDF15 hormone, has not been previously reported, and exhibits similar anorexigenic bioactivity to the canonical GDF15 hormone. The relative physiologic contribution of these two polypeptide products, CAP-GDF15 and canonical GDF15 hormone, from the same full-length polypeptide product remains unknown at this time. In addition, because the sequences are largely distinct, we suspect that the downstream receptor(s) of CAP-GDF15 are likely to be distinct from that of the canonical GDF15 hormone.”

Specific comments

1) In-silico analysis

It is not clear what the basis was for using GKK or GKR as the in-silico criterium for identifying C-terminally amidated peptides. There are several known bioactive peptides with C-terminal amidation formed from a GRR motif, including GLP-1, CCK8 and Orexin-B. The manuscript does not provide any argument for excluding GRR, which clearly also works with the PAM enzyme. I think the authors should comment on or justify their choice of motif here.

We apologize for this oversight and typo. In fact, {GKR/GRR} were used in the in-silico search. We have modified the text in the Results and the Methods: “Next, C-terminal amidation motifs were identified based on a GKR or GRR sequence indicative of dibasic cleavage and then amidation.” As the reviewer correctly notes, GKR and GRR are indeed much more prevalent glycine-dibasic residues compared to GKK at the C-terminus of known signaling peptides.

What is the argument for restricting the search to peptides of length 3-20 aa when many of the well-known and characterized bioactive peptides are longer than this? The manuscript should provide some justification or explanation for this seemingly arbitrary choice of length cut-off.

We agree that a justification for the search length should be provided. We have added the following **new text** to the Results: “Our search criteria were designed to use genome sequence information to “re-discover” known N-pyroglutamyl/C-amide peptides, and also uncover any additional potential peptides that, by sequence, might also contain similar chemical capping motifs.” In particular, we were inspired by the sequences of TRH (3-mer), GnRH (10-mer), and gastrin (17-mer), which are known N-pyroglutamyl/C-amide peptides that fall within this 3-20 amino acid range.

We also agree with this reviewer that additional capped peptides beyond the 20 aa limit could be in principle identified using this strategy. Notably, a key experimental limitation is in the chemical synthesis of authentic peptide standards, which becomes increasingly difficult with longer peptide lengths. Here, 20 amino acids was selected because this was the upper length where we could guarantee chemical synthesis of all peptide standards. To address this, we have added the following **new text** in the Discussion: “In addition, our peptide search space here was limited to those < 20 amino acids in length; studies of capped peptides > 20 amino acids in length is certainly warranted and possible provided that chemical synthesis of peptide standards can be accomplished.”

2) Is the number of in-silico identified peptides surprising or not?

The category “capped peptides” is a term invented by the authors and I see no problem with that as such. But the authors make several comments in the manuscript to indicate that the number of “capped peptides” identified is in their in-silico analysis and subsequent plasma analysis are somehow greater than what should be expected or what has previously been appreciated. It would strengthen the manuscript with some additional analysis substantiating that these sequences are more prevalent than what would be expected by randomly selecting sequences from all proteins, from secreted proteins, etc. using similar criteria. For instance, are there more sequences that have a Q within 3-20 amino acids of a GKK/GKR motif, compared to a V, T, M, etc.? The authors present it as a surprise that there are 216 sequences which live up to their criteria among thousands of secreted proteins. But is that a surprise at all? What would be expected by random sampling, taking the amino acid composition of the secreted peptides into account?

We thank the reviewer for providing an opportunity to clarify this point. Our intention was to compare the dozens of newly detected capped peptides with the much smaller subset of known peptide hormones/neuropeptides (e.g., < 5) that had been previously described in the literature to harbor co-incident N-pyroglutamylation and C-amidation. To clarify this comparison, we have added the following **new text** to the Results: “In addition, we establish that specific proteolytic processing and capping to produce protected peptide fragments is much more prevalent than previously anticipated, and certainly extends beyond the known subset of neuropeptides and peptide hormones containing these modifications.”

We also thank the reviewer for the suggestion to examine computational prediction of [X... GKR/GRR] sequences. We have performed these analyses, and find that in this sense, X=Q is not particularly enriched over other amino acids. We have included this in a **new Fig. S2C**, and added the following new text to the Results: “Changing the prediction criteria for the N-terminus to other, non-Q amino acids also produced ~50-300 predicted peptides (**Fig. S2C**), suggesting that our computational strategy may be even more general beyond co-incident N-pyroglutamyl/C-amide motifs.”

3) Detection of peptides in plasma is also presented as a surprise

The manuscript also presents the detection of roughly 30% of the in-silico identified peptides in plasma as surprisingly high and as a significant discovery that substantiates the claim that this group of peptides is something special and novel. This reviewer recognizes that the data looks solid and represents a valuable scientific contribution that demonstrates the existence in blood of these peptides. But this does not necessarily mean that their existence is highly surprising or that they are all functional bioactive peptides. That aspect should be discussed more by the authors.

Firstly, many different peptidomics studies have over the past decade reported tens of thousands of distinct peptide sequences that can be detected in mammalian tissues and blood. Most of these

are believed to be degradation products of mature proteins or bioactive peptides, aka not functional. Looking at the “capped” peptides that the authors highlight as novel discoveries, many of them are indeed shorter versions of peptides already known to exist. The authors should therefore discuss the possibility that many of the capped peptides detected in mouse (64) and human (85) blood could simply be degradation fragments of other peptides or proteins.

We agree with this limitation, and have now added the following **new text** to the Discussion: “Beyond these two specific examples, a major future challenge and goal will be to annotate the signaling and potential functions for other capped peptides. It may be possible that a subset of capped peptides may simply be degradation fragments from other proteins, and consequently non-functional.”

Secondly, one should expect the class of capped peptides to be more stable than other peptides in plasma (and hence enriched over randomly selected sequences) because they – by definition – harbor a C-terminal amidation which is known to stabilize peptides. So, even if these sequences were inactive fragments of bioactive peptides, they would likely still be stable enough to be detected in blood plasma. An example of this is the inactive GLP1(9-36Am) form of GLP1 which is formed when DPP4 cleaves the N-terminal end of the active GLP1(7-36Am)). Both can be detected in blood and tissues, as can shorter degraded versions. But only the full length 9-36 variant activates the receptor.

I therefore think they authors should discuss and argue more convincingly for why we should be surprised to detect 30% of the in-silico sequences in blood and be open to the possibility that many of the “capped peptides” may be non-functional degradation products.

While certain capped peptides are indeed functional and bioactive (e.g., CAP-TAC1 and CAP-GDF15), we agree that our data do not exclude the possibility that certain others may simply be non-functional degradation products. Indeed, in ongoing work we are actively attempting to define more rigorously and systematically the fraction of potentially active capped peptides using large-scale functional screening efforts. To clarify this point, we have **edited text** in the Discussion: “Lastly, large-scale screening of capped peptides against a panel of candidate G-protein coupled receptors, or via functional in vitro assays may also define the fraction of bioactive capped peptides versus those that are simply inert.”

Lastly, I would suggest a comparison of the sequences identified in plasma by the authors with peptides detected in other large peptidomics screens in tissues and blood, to show the reader if these peptides are novel discoveries in the sense that they have never been reported before (with and without the pyro-Glu).

We agree, and have now directly compared the sequences (with and without the pyro-Glu) reported here with the public database PeptideAtlas. None are found in the Peptide Atlas Mouse Plasma Build (with or without modification). We interpret the lack of overlap to reflect the fact that large-scale datasets are identifying the most abundant peptides and peptide fragments, whereas our targeted approach enriches for lower abundance peptides. We have added the following **new text** to the Results: “Lastly, the capped peptide sequences identified here are not found in PeptideAtlas, which may be attributable to the higher sensitivity, targeted mass spectrometry approach used for their detection here compared to shotgun approaches.”

4) Novelty of the CAP-TAC1 case story

To my surprise, the CAP-TAC1 findings are highlighted as the main case-story in the manuscript

where it is presented as a novel peptide which binds to TACR1/NK1R. For instance, one sentence in the discussion reads “Interestingly, CAP-TAC1 is only 5 amino acids in length and therefore represents the shortest naturally-occurring tachykinin receptor agonist reported to date in any species.”

This reviewer finds the wording a bit inflated and the novelty a bit overstated, considering that many other shorter versions are known to be bioactive, including a sequence identical to CAP-TAC1, just without the pyro-Glu.

Here, I think the authors should present the data in a way that makes it easier for the reader to assess the novelty and impact of their discoveries in an area where there is already a lot of prior work demonstrating activity of shorter variants of SubstanceP.

The novel CAP-TAC1 sequence is pGlu-FFGLM-Am which corresponds exactly to the C-terminal end of Substance P. It also has very high similarity to the C-terminal ends of Neurokinin A and B.

Substance P (TAC1 derived): RPKPQQFFGLM-Am
Neurokinin A (TAC1 derived): HKTDSFVGLM-Am
Neurokinin B (TAC3 derived): DMHDFVGLM-Am

The fact that this shorter sequence identified by the authors (CAP-TAC1) has affinity for the same receptors as SubstanceP is not that surprising to this reviewer. Especially, considering that previous work (cited by the authors) has shown that the non-pyro version of their “novel” CAP-TAC1 peptide (QFFGLM-Am) is a potent agonist of the tachykinin receptors and that other highly similar sequences such as FGLM-Am and pGlu-FFPLM-Am also work.

Given this data, the authors should provide a figure showing the many highly similar sequences already known to be bioactive and demonstrate that their pyro-Glu variant is different in terms of activity to the non-pyro-Glu variant which has already been shown to be active too.

We entirely agree with this reviewers’ suggestions.

First, we have removed the sentence, “Interestingly... reported to date in any species.”

Second, we have added a **new panel (Fig. 4B)** showing the similar sequences between CAP-TAC1 and the other tachykinins and tachykinin fragments. We have **modified the following text** in the Results: “We noted that the sequence of CAP-TAC1 contains the key consensus C-terminal FXGLM motif which is characteristic of all known tachykinin neuropeptides (**Fig. 4A,B**). In addition, the C-terminal methionyl amide, which is also present in CAP-TAC1, had previously been shown to be critical for agonist activity of other tachykinin neuropeptides.”

Substance P	-----RPKPQQFFGLM-NH2
CAP-TAC1	-----pGlu-FFGLM-NH2
Substance P (6-11)	-----QFFGLM-NH2
Neuropeptide gama	DAGHGQISHKRHKTDSFVGLM-NH2
Neurokinin A	-----HKTDSFVGLM-NH2
Neurokinin B	-----DMHDFVGLM-NH2

Third, we used the receptor agonist activity assay to determine potential functional differences between the previously reported substance P fragment (QFFGLM-NH2) and CAP-TAC1 (pGlu-FFGLM-NH2). As shown in a **revised Fig. 4**, our data shows that N-pyroglutamylation confers significant functional differences in terms of receptor engagement and stability compared to the

unmodified N terminus (e.g., Substance P[6-11]). We have now added the following **new text** to the Results: “A C-terminal fragment of substance P (amino acids 6-11, QFFGLM-NH₂) had been previously reported to be an endogenous peptide and also an agonist at the tachykinin receptors. Substance P(6-11) differs from CAP-TAC1 in that its N-terminus is unmodified (e.g., not cyclized), but the remainder of the sequence is otherwise the same. Using the same TACR1 agonist assay, we found that Substance P(6-11) was approximately 2-fold less potent and also exhibited reduced maximal response compared to CAP-TAC1. [...] Using similar cellular agonist assays for TACR2, we found that CAP-TAC1 exhibited 3-fold higher potency than the control Substance P; in addition, for this receptor, CAP-TAC1 and Substance P(6-11) were largely functionally indistinguishable. For TACR3, CAP-TAC1 was 27-fold more potent than both full-length Substance P and Substance P(6-11). For this receptor, CAP-TAC1 also exhibited a ~20% higher maximal activation compared to Substance P(6-11). We conclude that CAP-TAC1 is a full agonist of multiple mammalian tachykinin receptors with potency similar to, or in some cases higher than, than previously established tachykinin neuropeptides. In addition, our data demonstrate that N-terminal pyroglutamylation confers functional differences in receptor engagement compared to the unmodified N terminus.”

5) Are “capped peptides” a novel class of peptides?

The two modifications (N-term PyroGlu and C-term amidation) are both well known to 1) be formed in vivo (enzymes exist for creating both) 2) to be important for stability and function. The novel idea here should therefore be that the combination constitutes a special class of endogenous peptides that has hitherto been overlooked.

I think the authors should argue a bit better for this point, or alternatively revise the wording in their manuscript. Many of the sequences seem to just be shorter versions of already known peptides with a C-terminal amidation. What is lacking is a demonstration of two things:

A) the uniqueness of this particular shorter variant. It would strengthen the manuscript with a few

examples where the authors demonstrate experimentally that slightly longer or slightly shorter versions of these peptides do not exist in vivo, aka that only the variant which start with a Q is present in blood. A lot of previous peptidomics papers have found “ladders of degradation” where pretty much all shorter fragments exist in-vivo, pointing to a degradation mechanism where, for instance, the N-terminal is gradually degraded. Since the authors only look for one variant with their targeted approach, they are “blind” to the potential existence of other fragments that may be equally present and/or active.

B) that the pyro-Glu variant is indeed more stable and more bioactive than the “raw” Q variant, and more stable/active than slightly longer or shorter sequences. Unless “capped peptides” stand out distinctly from other related variants in terms of abundance and bioactivity then there is no strong basis for claiming that these represent a special, biologically relevant sub-category of peptides that others in the field should be interested in.

As the authors point out themselves, at least 3 capped peptides are already well characterized, so the conceptual idea that a double-capped peptide with pyro-Glu and Amidation exist is not novel. The novelty claimed here is that they are more abundant, more important, etc. than previously recognized, and this reviewer does not think that the manuscripts argues sufficiently strong for that.

We agree and have made the following modifications in response to this point.

First, we agree with this reviewer’s main point that the wording of the manuscript can be revised. We have added the word “potential” and also removed any word of “orphan” to describe the class of capped peptides. For example, the abstract now reads: “Here we demonstrate the endogenous presence of a sequence diverse class of blood-borne peptides that we call ‘capped peptides.’ ”

In addition, we have now performed **new experiments** to experimentally address reviewer’s specific inquiry about the comparison of the pyro-Glu versus the “raw” Q variant for both CAP-TAC1 and CAP-GDF15. As described in response to reviewer question #4 and in a **revised Fig. 4**, we have already shown that CAP-TAC1 (pyroGlu) and Substance P(6-11) (“raw” Q) exhibit differential agonism of the mammalian tachykinin receptors. In addition, CAP-TAC1 has a different plasma stability compared with Substance P(6-11). We have now performed new experiments that compare CAP-GDF15 with an “uncapped” CAP-GDF15 (e.g., pGlu-LELRLRVAAGR-NH₂ versus QLELRLRVAAGR-COOH). These data are shown in **Fig. S4F**. To describe these data, we have added the following **new text** to the Results: “Additionally, we synthesized an uncapped version of CAP-GDF15 (no pyro-Glu or amidation, QLELRLRVAAGR-OH) and found this uncapped version to induce significantly less of food intake reduction compared to the fully capped CAP-GDF15 (**Fig. S4F**).”

6) The method works for finding bioactive peptides...or at least for finding one exciting example

Having said all of the above, the method does identify the GDF15-derived peptides which seems to be novel both in terms of sequence, detection and function. The fact that this peptide is NOT a sub-fragment of the GDF15 hormone itself but rather a C-terminal fragment of what is currently believed to be a pro-peptide of the same precursor make this a very interesting finding. In that sense, the GDF15 story is much more novel and surprising than the CAP-TAC1 story. The fact that this new peptide reduces food intake and lowers body weight in-vivo makes it a discovery worthy of publication in itself. It would suggest to give this story a more prominent place in the manuscript, since this – to this reviewer – represents the most significant scientific discovery in this work.

We agree, and have therefore added the following sentence to the abstract to emphasize this point. For example, in the abstract: “A second capped peptide, CAP-GDF15, is a previously unknown 12-mer peptide cleaved from the prepropeptide region of full-length GDF15 that, like the canonical GDF15 hormone, also reduces food intake and body weight.”

As mentioned in response to this reviewer’s overall comments, we have also added **new text** to the discussion to emphasize CAP-GDF15: “Like CAP-TAC1, the detection of CAP-GDF15 also demonstrates that a single full-length preproprecursor (in this case, full-length GDF15) can generate more than a single bioactive polypeptide product. Notably, CAP-GDF15 is novel in the sense that it is not part of the canonical GDF15 hormone, has not been previously reported, and exhibits similar anorexigenic bioactivity to the canonical GDF15 hormone. The relative physiologic contribution of these two polypeptide products, CAP-GDF15 and canonical GDF15 hormone, from the same full-length polypeptide product remains unknown at this time. In addition, because the sequences are largely distinct, we suspect that the downstream receptor(s) of CAP-GDF15 are likely to be distinct from that of the canonical GDF15 hormone.”

7) Formation of pyro-Glu

It is generally assumed in the manuscript that because the pyro-Glu peptides are detected in blood, they also exist in-vivo. This is not an unreasonable assumption given that (in humans) the pyro-Glu conversion is catalyzed by two isoforms of Glutaminyl cyclases (<https://www.sciencedirect.com/science/article/pii/S0014299922004393>), one secreted and one located in the Golgi.

Unless the conversion is spontaneous, these enzyme must come into proximity of the proposed capped peptides, which could occur both in the secretory pathway or extracellularly.

However, spontaneous pyro-Glu formation has been described and discussed in the literature and has formation as an experimental artifact. How sure are the authors that the detected pyro-Glu capped peptides are in fact present in-vivo and not an artifact of the experimental analysis? There are reports which seem to indicate that these modification can occur as experimental artifacts: <https://pubs.acs.org/doi/10.1021/ac501451v> and <https://pubs.acs.org/doi/10.1021/acso mega.9b04384>. It would strengthen the manuscript with some additional arguments for the in-vivo relevance and/or more discussion of the possibility of these findings being artifacts.

We thank the reviewer for this valuable suggestion. We have now generated **new data** to exclude the possibility of artefactual formation of pyroglutamate, which are presented in a **new Fig. S2A**. We have added the following **new text** to describe these data: “To exclude the possibility that N-

pyroglutamylation may be artefactually occurring in the sample preparation, we subjected a synthetic standard of uncapped CAP-TAC1 (QFFGLM) to the sample preparation conditions. As shown in **Fig. S2A**, we did not observe any formation of pGlu-FFGLM from the QFFGLM starting material.”

N- or C-cap formation with peptidomics preparation of uncapped CAP-TAC1

In the context of this discussion, it would again be relevant for the authors to demonstrate that the pyro-Glu variant is indeed more stable or more active than its non-modified counterpart and/or that slightly longer or shorter variants are not stable and active too. That would strengthen the case for the functional importance of the pyro-Glu modification and rule out the possibility that the stability and function is mainly or solely driven by the amidation.

We thank the reviewer for suggesting this experiment, which we have now performed and included in **a revised Fig. 4F**. We have added the following **new text** to the Results: “Beyond differences in TACR activation, we reasoned that the two chemical caps of CAP-TAC1 might also produce important functional differences in terms of stability and resistance to proteolytic degradation compared to Substance P. To directly test this possibility, CAP-TAC1 (10 μ M) and substance P (10 μ M) were individually incubated with mouse plasma and incubated at 37°C. and their levels over time were measured by LC-MS. Substance P exhibited time-dependent degradation with a $t_{1/2}$ = 8.4 min. By contrast, the rate of CAP-TAC1 degradation was substantially slower ($t_{1/2}$ = 14.4 min) (**Fig. 4F**). In fact, levels of CAP-TAC1 were still detectable after 90 min, a time point when Substance P was undetectable (**Fig. 4F**). Substance P(6-11) also exhibited rapid degradation kinetics which were distinct from CAP-TAC1 (**Fig. 4F**).”

Reviewer #3 (Remarks to the Author):

The manuscript describes the use of bioinformatics to predict a specific class of peptide that has a C-terminal amide and an N-terminal pyroglutamate. The authors then synthesised a large

number of the theoretical peptides and then assessed if they were present in the circulatory peptidome of mice. Whilst this is a possible avenue for detecting and identifying bioactive peptides, I am concerned that the peptides they have selected for further study are not likely to be produced endogenously. I have highlighted some areas of concern I have with the manuscript.

Peptide selection, and potential route of endogenous production. Most bioactive peptides in mammalian systems have basic amino acids present in or near their theoretical cleavage sites. Whilst the reviewers do discuss this, the main two peptides they discuss are cleaved from their prohormone sequences without these basic amino acids immediately upstream of the N-terminal – or in the case of CAP-TAC1, there are basic amino acids two residues upstream from the proposed cleavage site.

The CAP-TAC1 peptide from the mouse described in the paper – shown below seems to have two amino acids upstream on the N-terminus in front of the proposed n-terminal pyroglutamate of the peptide (pQFFGLM-NH₂). This peptide could still possibly be present, however would need to be produced after the original PQQFFGLM peptide was produced and subsequently cleaved by an enzyme such as DPP-IV. Or substance P itself was produced and had four amino acids removed enzymatically. The same sequence is obviously present in human – with the additional PQ on the N-terminus of the proposed peptide sequence.

Mouse TAC1 sequence:

MKILVALAVFFLVSTQLFAEEIGANDDLNYWSDWYDSDQIKEELPEPFEHLLQRIARRPKPQQFF
GLMGKRDADSSIEKQVALLKALYGHGQISHKRHKTDSEVGLMGKRALNSVAYERSAMQNYER
RR

Human sequence:

MKILVALAVFFLVSTQLFAEEIGANDDLNYWSDWYDSDQIKEELPEPFEHLLQRIARRPKPQQFF
GLMGKRDADSSIEKQVALLKALYGHGQISHKRHKTDSEVGLMGKRALNSVAYERSAMQNYER
RR

The same issue is apparent for the GDF15 sequences that are described in the manuscript. The mouse GDF15 sequence for CAP-GDF15 doesn't have any sequence for the peptide to be cleaved from the prohormone sequence upstream of the initial N-terminal pyroglutamate of CAP-GDF15, and the human sequence has a RP sequence immediately upstream, which is unlikely to result in a preferred cleavage site by standard prohormone convertases. Ghrelin has a PR sequence at the C-terminus, which results in its cleavage from the prohormone sequence, but a RP is unlikely to be cleaved.

Human GDF-15 sequence

MPGQELRTVNGSQMLLVLLVLSWLPHGALSLAEASRASFPGPSELHSEDSRFRELKRYEDL
LTRLRANQSWEDSNTDLVPAPAVRILTPEVRLGSGGHLHLRISRALPEGLPEASRLHRALFRL
SPTASRSWDVTRPLRRQLSLARPQAPALHLRLSPPPSQSDQLLAESSARP**QLELHLRPQAA**
RGRRRARARNGDHCPLGPGRCRLHTVRASLEDLGWADWVLSPREVQVTMCIGACPSQFRA
ANMHAQIKTSLHRLKPDTPAPCCVPASYNPMVLIQKTDGTVSLQTYDDLLAKDCHCI

Mouse GDF-15 sequence

MAPPALQAQPPGGSQLRFLFLLLLLLLLLSWPSQGDALAMPEQRPSGPESQLNADELGRFQ

DLLSRLHANQSREDSNSEPSPDPAVRILSPEVRLGSHGQLLLRVNRSLSQGLPEAYRVHRAL
LLLTPTARPWDITRPLKRALSLRGPAPALRLRLTPPPDLAMLPSGGT**QLELRLRVAAGR**GRR
SAHAHPRDSCPLGPGRCCHLETVQATLEDLGWSDWVLSRQLQLSMCVGECPHLYRSANTH
AQIKARLHGLQPKVPAPCCVPSSYTPVVLHVRTDSGVSLQTYDDLVAR**GCHCA**

As can be seen from the above mouse GDF15 sequence, there are no basic amino acids upstream of the proposed peptide sequence, therefore the peptide is unlikely to be formed as part of the normal bioactive peptide cleavage process. The authors need to discuss the proposed cleavage mechanism for this peptide from the GDF15 precursor sequence. The authors should look at the cleavage positions of the other peptides they discuss (such as GDNF, FGF5, PLA2G2A) and discuss how these peptides would be produced endogenously.

We agree with this reviewer's suggestion that additional data/discussion should be added to discuss potential cleavage mechanism for liberation of the N-terminus of capped peptides. We have now performed **new experiments** and added **new text** to address this point.

First, we re-visited the in vitro experiments where we measured Substance P half-life in mouse plasma. We now show that in these assays we can observe the biochemical formation of CAP-TAC1. These **new data** are shown in **Fig. S3** and provide experimental demonstration of this reviewer's hypothesis that CAP-TAC1 production may involve exopeptidase processing of full-length Substance P. We have added the following **new text** to the Results: "Lastly, we also observed that both Substance P(6-11) and CAP-TAC1 were formed upon degradation of full-length substance P, suggesting that a combination of exopeptidase and glutamyl cyclase activity together can generate these smaller two peptides (**Fig. S3**)."

Second, we agree that additional discussion should be provided about the proposed cleavage mechanism for CAP-GDF15 and other capped peptide examples. We have therefore added the following **new text** to the Discussion: "The proteolysis pathways leading to the production of capped peptides remains an important area for future work. While classical peptide hormones and neuropeptides are liberated from their preproprecursors via the action of proprotein convertases, many of the capped peptides that we detect lack an immediate upstream dibasic residue. One possibility, which we experimentally demonstrated for CAP-TAC1, is that proprotein convertase first cleave at a dibasic site further upstream of the N-terminal pyroglutamyl residue, and the resulting (longer) peptide is then trimmed via exopeptidase activity and the N-terminal pyroglutamate is subsequently installed (...**RRPKPQQFFGLMGKR**...). Such a biogenesis mechanism may also contribute to the production of CAP-GDNF (...**RRERNRQAAAASPENSRGKGR**...). Others lack a proximal upstream dibasic residue; for these, we speculate that proprotein convertase-independent proteolytic mechanisms may be operational. For instance, CAP-PLA2G2A is found several amino acids C-terminal to the signal

peptide of full-length PLA2G2A (signal peptide, amino acids 1-21; CAP-PLA2G2A amino acids 25-35); consequently, its N-terminus might be liberated via sequential signal peptidase and exopeptidase activity. Lastly, for peptides including CAP-GDF15, we speculate additional proteases might be involved in liberation of the N-terminus. For instance, cathepsin L has been reported to be involved in the biogenesis of other peptide hormones/neuropeptides and exhibits very broad substrate specificity beyond basic residues alone.^{21,22} Interestingly, cathepsin L and GDF15 are both highly expressed in macrophages. Moreover, proteomic studies of neo-N-termini from extracellular proteins have revealed a diversity of neo-N-terminal amino acids;²³ such non-canonical proteolysis pathways may be operational in the production of capped peptides.”

With regards to the supplied evidence of confirmation of detected endogenously produced peptides, I feel that this needs to be strengthened significantly.

pQFFGLM-NH2 peptide supplied data.

The supplied images in figure 1 (panel E and F) and S1 do not look like raw data exported from an MS software, however looks like the data has been reproduced in a different program. This should be re-done to include original data. The spectra in figure 1F only shows 1 decimal place – and obviously this value has been rounded. Current mass spectrometers – including the one used by the authors – are capable of much higher accuracy. The MS/MS spectra of the peptides detected in an in-vivo setting and their in-vitro generated equivalents should be reproduced in much clearer fashion, so that the readers can definitely see that the peptides match.

pQLELRLRVAAGR-NH2 peptide supplied data.

The supplied images in figure 5 (panel B and C) again does not look like raw data exported from an MS software. This should be re-done to include original data. Again - the spectra only shows 1 decimal place. It seems that there only MS/MS spectral data is displayed up to the precursor m/z value, was data not acquired above this the precursor m/z value? – this would give more data for comparing the endogenous and synthetic peptides. If the authors are claiming to detect these peptides endogenously, more data should be provided – such as demonstrating that the precursor ions match exactly, proper chromatographic comparison and product ion spectral matching. This data should not be retouched using a separate program, but should come directly from the MS software packages used to look at the data.

We have added substantial **new experiments**, to support the endogenous detection of capped peptides.

First, we have provided in a new **Fig. S1** (mouse) and **Fig. S5** (human) the raw chromatographic traces from the LC-QTOF instrument showing elution times and high-resolution masses for the MS1 of the authentic peptide standard and endogenous peak. Importantly, this addresses the reviewer’s main questions, including showing higher mass resolution (beyond 1 decimal point) as well as raw unprocessed chromatograms (e.g., screenshots of the Agilent MassHunter Software).

Second, we have performed new targeted measurements for each detected capped peptide using multiple-reaction-monitoring on an LC-QQQ instrument. Notably, these methods enable (at unit mass) isolation, fragmentation, and measurement of specific parent-to-daughter transitions and provide important independent validation of our original LC-QTOF data. We have elected to use the LC-QQQ approach as secondary confirmation (rather than showing all mirror plots) because of the significantly higher sensitivity of triple quadrupole analysis. In these LC-QQQ experiments, our positive detection criteria was defined by retention time (within 1 min), detection of a key transition identified from the authentic standard, and a signal-to-noise ratio > 2.5. The specific

transitions, collision energies, and fragmentor voltages are provided in **Table S2**. The raw data are again included in **Fig. S1** (mouse) and **Fig. S5** (human). A representative subset of these is included here, and we refer the reviewer to those Supplementary Figures for the remaining panels.

For clarity, and to emphasize the targeted LC-QQQ methods, we have now removed the mirror plots originally presented in Fig. 1 and 5.

Issues surrounding the dosage of the GDF15 derived peptide

Dosage of the GDF15 peptide to animals was performed at 50mg/kg – as long as this isn't a typographic error and should be 50µg/mL, this level of dosing is absolutely immense for a potentially bioactive peptide. Most bioactive peptides are dosed at far lower amounts in pre-clinical studies – usually in the 1-3 mg/kg range. The plasma levels of these dosed peptides would be in the very high µg/mL, which is significantly above what levels of the endogenous peptides would reach. GDF15 is present in mouse plasma at approximately 100 pg/mL – a quick google found this article <https://doi.org/10.1038/s41598-019-56922-w> the authors should indicate what the concentration of their CAP-GDF15 peptide reached in plasma and why it was dosed at such a high level. The authors need to discuss the reason for the large dose and highlight that this level would be significantly above any potential endogenous level.

We find that basal levels of CAP-GDF15 are in the range of ~1-10 nM or ~1-10 ng/ml (**Table S2**). As the reviewer notes, this is approximately 10- to 100-fold higher concentration reported for the canonical GDF15 hormone, which may reflect differences between CAP-GDF15 and the canonical GDF15 hormone in a combination of stability, clearance, or potentially interactions (e.g., complexes) with other plasma proteins. Notably, similar magnitude differences in levels for peptides derived from the same precursor have been previously reported (e.g., iC3B is 100-fold more abundant than C3a; 5 ug/ml versus 50 ng/ml, respectively; PMIDs 25660530 and

32973774). Following IP dosing at 50 mg/kg, we do not find evidence that CAP-GDF15 levels rise to ug/ml concentrations; rather, our measured levels of CAP-GDF15 are increased by ~5-fold after administration. These **new data** are now shown in **Fig. S4A**. We have added the following **new text** to the Results: “At this dose, concentrations of plasma CAP-GDF15 rose by 5-fold from baseline at 30 min post-administration (**Fig. S4A**).”

In the future, it will be important to establish more accurate dose-response and PK/PD relationships for CAP-GDF15 and its anorexigenic activity. To address this point, we have added the following **new text** to the Discussion: “Lastly, it will be important in future studies to establish the full dose-response and pharmacokinetic/pharmacodynamic profile of CAP-GDF15.”

To confirm that these identified peptides are indeed produced endogenously, then the authors need to compare the concentration of the peptide they are identifying and compare these to the expected concentration in the animal. For example, endogenous levels of substance P is in the low tens of picograms per millilitre – what is the concentration of the level of the capped peptide that was detected? If these concentrations are comparable with the expected concentration of the more classical prohormone derived protein, then it would add weight to their potential as bioactive peptides.

We detected CAP-TAC1 at circulating concentrations of ~ 1 nM = ~ 700 pg/L. Concentrations of circulating substance P vary substantially in the literature and have been reported at 10-300 pg/L (PMID 16971517). Therefore the concentration of CAP-TAC1 appears to be within 3-fold of the high estimates of plasma Substance P levels. More generally, the concentration range all capped peptides detected is 100 pM – 100 nM. We have added the following **new text** in the Discussion: “The observed concentrations of capped peptides (100 pM to 100 nM) falls within the range circulating concentration range known signaling peptides, such as gastrin, glucagon, insulin, and leptin, which are also found in blood plasma at picomolar and nanomolar concentrations.”

Minor issues

The sensitivity that the authors have achieved for the peptide analysis is very good for a high flow analyses (0.7 mL/minute). Looking at the extraction methodology, it is unclear how much plasma was extracted – was it 100 µL or 1mL – please clarify.

We have now **edited text** the Methods for clarity: “Protease inhibitor (HALT, ThermoFisher, #78429) was added to plasma (10 uL HALT into 1 mL plasma). Plasma was diluted 1:6 plasma:Tris-HCl buffer (100 mM Tris-HCl, pH 8.2) and boiled at 95°C for 10 minutes. In total, 1 ml of pooled plasma was used per replicate.”

REVIEWER COMMENTS

Reviewer #1 (Remarks to the Author):

The authors have answered most of the reviewer's questions. However, there is one serious issue regarding of the identification criteria of the capped peptides. Since the majority of this research work represent a data resource of the identified new peptides, the manuscript cannot be accepted without further solid experimental evidences on that. Please see the details on this issue as shown below.

1. Corresponding to Q1, Q4 of Major issues raised by Reviewer #1.

In the previous version of the manuscript, the authors show the mirror matching of MS/MS spectra between synthetic standards and plasma endogenous peptides (Figure 1F, 5C in the previous version). This represents one of the golden standards for identification of new/unknown peptides in biological samples using mass spectrometry. In the revised version, the authors removed this part and replaced it with a targeted MS using MRM as a previous report by Donohue et al. (PMID: 33360778). However, Donohue et al. used this MRM approach for quantitation of peptide hormones instead of identification. To identify new or unknown peptides, a perfect MS/MS matching between synthetic standards and endogenous peptides is necessary. Besides that, LC retention time matching is required to exclude the possibility of peptide's isomeric difference. We understand that the target peptides' abundance could be too low to obtain high-quality MS/MS spectra using regular flow LC-MS. However, without that, the false identifications of these capped peptides could mislead the scientific community. Therefore, please improve the peptide enrichment and MS detection approaches (perhaps targeted analysis on Orbitrap instruments using nano-flow LC) and obtain high quality MS/MS spectra.

For examples, the two peptides cannot be distinguished by using the authors' MRM approach: ABCD or BACD. The two peptides might have the same molecular weights, similar RT values, and the same b₂, b₃, y₁, y₂ ions, but they are different two peptides. The second example is ABCD or ABCDamide. The amidation only show 1 Dalton mass shift. The two peptides might have the similar RT value and the same b₁, b₂, b₃ ions, and all the y ions only show 1 Dalton difference. The mammalian's proteome is extremely complicated, and mis-matching often occurs. Therefore, mirror matching of MS/MS is necessary.

2. Corresponding to Q4 of Minor issues raised by Reviewer #1.

It is improper that the authors removed the problematic figure and avoided the reviewer's question. It is not difficult to annotate the MS/MS spectrum, because pEELRLRVAAGRamide is a regular peptide and should result in abundant fragment ions. Why delete that?

Overall, this reviewer suggests further experimental evidences on the MS/MS matching. Without that, the data resource is doubtful.

Reviewer #2 (Remarks to the Author):

Overall comments

The authors have done a good job addressing the points I raised, adding new experimental data, analyses and figures/tables. The manuscript represents an advance in our knowledge about peptides by demonstrating the existence in mouse and human blood of many short (3-20) peptides harboring the pyro-Glu / C-term Amidation combination in human and mouse blood. They show bioactivity for some of them and although their primary example (CAP-TAC1) is nearly identical in sequence to several peptides already known to be active (SubstanceP and neurokinins), they have added new data to demonstrate that the pyro-Glu modification leads to increase stability and a different receptor affinity profile. To this reviewer, the GDF15 peptide is still the main biological discovery because it is

both truly novel and bioactive.

Specific comments (relating to authors rebuttal)

Add 1) the in-silico analysis

I am glad that my question made the authors realize that they had in fact included GRR in their original analysis, not GKK. But this of course turns attention to the reason for leaving out GKK.

The authors provide the argument that this reviewer should have somehow claimed that GKR and GRR are more prevalent than GKK but I do not recall saying that anywhere in my review. Nor do I think it is necessarily true. A quick search reveals that in mouse, Gastrin-releasing peptide, Neuromedin-C, Neuromedin-B and Melanocyte-stimulating hormone alpha are all formed from GKK motifs where the G is converted by PAM into the C-terminal amidation. So, PAM certainly works on GKK too.

This is not a critical point, and the revised wording is ok, although it still seems that GKK was left out based on an assumption of not being as relevant as the two other motifs for C-terminal amidation, and that doing so may have left out potential candidate peptides from their in-silico analysis. But I acknowledge and accept that the authors can't redo the whole project to include GKK bearing peptides.

Regarding the question of why the length cut-off was set at 3-20 amino acids, I thank the authors for their honesty in the feedback provided to my comment. Based on that, I suggest that they revise the manuscript further to also let the readers know the real motivation for limiting the length to 20 aa (in the main text when they mention the 3-20 aa search criterium).

The changes made in response to my question is addition of some rather cryptic new wording added about "re-discovering known peptides" plus new comments in the discussion.

I would much prefer if they just said in the main text that they chose 20 aa as the length cut-off because this is the upper limit of their chemical synthesis capabilities and because the only 3 known peptides with the capped peptide characteristics are all shorter than 20 aa. This is a perfectly valid argument for the choices they made in their analysis, and I see no need to "hide" this from the reader.

Add 2) Is the number of in-silico identified peptides surprising or not?

The authors have performed the analysis I suggested, and although it is not backed by statistical analyses, it appears – as the authors point out themselves - to show that there is nothing unusual or unexpected about the frequency of peptides with a Q compared to other amino acids. They are no more prevalent in the proteome than would be expected statistically.

This finding would perhaps be worthy of a few comments and reflections, since one would normally expect a higher level of evolutionary conservation from motifs and modifications which play a vital biological role.

In general, I think the authors should consider if they can somehow avoid presenting their in-silico findings as a major surprise in terms of the number of sequences identified (without backing this with evidence), since the evidence suggest that the number of peptides identified in-silico are probably no larger than for any other amino acid than Q.

What the authors have unfortunately done instead in the revised manuscript, is to use the new analysis (of the N-terminal amino acid frequencies) to conclude that their in-silico strategy is worth

extending beyond pyro-Gly/C-am.

To me, this is a kind of nonsense argument. If one looks for the occurrence of any given amino acid 3-20 positions upstream of a GKR/GRR motif in a protein, one is bound to find many examples, and the authors do. But that does not necessarily make this an interesting analysis worth pursuing for peptides with alternative N-terminal residues. The argument also contradicts their own key point, namely that pyro-Glu in combination with C-terminal amidation is something special. Their work indicate that it might well be, so I would omit this comment, as it only adds confusion.

The first new sentence added in their response could also be sharpened to just say that their results demonstrate the existence of many new examples of "capped peptides" detectable in blood. That is interesting in its own right.

I would, however, suggest to leave out all the (unsubstantiated) comments about the number of peptides discovered in-silico or experimentally being much larger than expected or anticipated, unless the authors can provide references to previous work which contains estimates or speculation on the number of pQ/C-am peptides. To my knowledge, there are no references speculating about the potential true number of this particular combination of modifications, and it is therefore unnecessary to claim that this new work finds more than were previously anticipated or expected.

Add 3) Detection of peptides in plasma

The authors have addressed most of the comments, and my only remaining feedback is around the novelty of the identified sequences.

Comparing to PeptideAtlas is a nice addition but it may also be worth considering comparison to other large-scale peptidomics screens. For instance, QMAVKKYLNSILN (derived from VIP) was reported in <https://www.ncbi.nlm.nih.gov/pmc/articles/PMC4857386/> in 2016 (including the amidation), and some of the other sequences reported by the authors as novel were reported to be detectable in whole tissues by Madsen et al. <https://www.nature.com/articles/s41467-022-34031-z>). This strengthens the case for their existence and relevance in-vivo.

The authors could also highlight that their approach has an advantage in the ability to identify very short peptides, compared to MS-based peptidomics screens which usually do not search for peptides shorter than 7-8 aa.

As mentioned earlier, it is not surprising to this review that it is possible to detect sequences which are shorter versions of known bioactive peptides, given all the previous work which has demonstrated the existence of many shorter versions in-vivo. It is thus not the existence of "capped" peptides that is the main story but the fact that at least some of them appear to be both stable and bioactive.

Add 4) Novelty of the CAP-TAC1 case story

Very nice additions, figures and data which makes it easier for the reader to understand the many highly related sequences while now also demonstrating that the pyro-Glu actually matters to the activity of the peptide. No further suggestions on this from my side.

Add 5) Are "capped peptides" a novel class of peptides?

On this point, the authors have added very nice new data and figures which now provide much better evidence for the pyro-Glu versions being functionally distinct from unmodified variants.

Add 6) The method works for finding bioactive peptides

I very much applaud the changes here which shine additional light on the truly novel bioactive GDF15 peptide discovered by the authors. The fact that CAP-TAC1 is active against the same receptors as Substance P is not surprising given that CAP-TAC1 is a sub-sequence of Substance P with a pyro-Glu modification. But demonstrating that the novel GDF15 peptide also lowers appetite is a significant discovery because this peptide is new and non-overlapping with the well known GDF15 peptide. The observation that the RR site which is cleaved to form the classical GDF15 (MIC1) peptide has an upstream G support the case for conversion of this glycine to an amidation that the authors detect.

Add 7) Formation of pyro-Glu

No further comments on this part, following the new additions which provide evidence against the idea that the pyro-Glu modifications could be experimental artifacts, and new data demonstrating differences between Glu and pyro-Glu versions of the same peptide.

Reviewer #3 (Remarks to the Author):

The authors have taken into account my previous comments, and I am happy with most of the adjustments.

I however still have an issue with the administration of the GDF15 peptide to the mouse. The authors dose a mouse at 50 mg/kg, therefore taking a conservative value of a mouse being 30g, approximately 1.5 mg of the peptide was dosed in one animal.

According to figure S4A, this caused an increase of the endogenous peptide level from approximately 1 nM to a maximum of 4 nM. A concentration of 4 nM is equivalent to around 5 ng/mL for the endogenous GDF15 peptide, therefore the blood concentration was raised from around 1 to 5 ng/mL after the dosing of 1.5 mg of peptide.

Whilst there isn't enough data to work out the area under curve of the administration, this does suggest that the bioavailability of the peptide via an I.P route administration is incredibly poor.

For example, comparing the data from Liraglutide administration (<https://doi.org/10.2337/diacare.25.8.1398>) in humans at 5 µg/kg (10,000 times lower dosing than this study), equating to approximately 0.4 mg peptide dosage to a human based on 80kg weight (3x less overall peptide dosed than this study – but to a human). This liraglutide administration to humans returned a similar CMax of 4.5 nM, and with a bioavailability of 55%. See data in table 1.

If the administration details of the GDF15 peptide to mice are to be included, the low bioavailability of the peptide must be stated.

Reasons for this low plasma value could be that potentially the peptide was being lost quickly – either through tissue distribution or loss through filtration. I am assuming not through degradation due to the peptide being capped at both ends.

The reference ranges of the peptide hormones glucagon and gastrin suggested by the authors are not quite as high as stated. Normal glucagon and gastrin ranges are 1-100 pg/mL which corresponds to ~30 and 50 pM of glucagon and gastrin respectively for the upper limit of the ranges. Therefore the authors statement that CAP peptides are circulating around the level of the canonical gastrin and glucagon peptides is incorrect and these two peptides should be removed from the statement.

Minor comments

1. Parent-daughter nomenclature should really be changed to precursor-product throughout.
2. Please make sure m/z is italicised throughout.
3. Fig S3 has "from" duplicated in the title
4. As stated in the revised document "At this dose, concentrations of plasma CAP-GDF15 rose by 5-

fold from baseline at 30 min post-administration (Fig S4a)". The S4A figure shows that the 10 minute time point post administration has the highest CAP-GDF15 peptide level. Please correct the text.

REVIEWER COMMENTS

Reviewer #1 (Remarks to the Author):

The authors have answered most of the reviewer's questions. However, there is one serious issue regarding the identification criteria of the capped peptides. Since the majority of this research work represent a data resource of the identified new peptides, the manuscript cannot be accepted without further solid experimental evidences on that. Please see the details on this issue as shown below.

1. Corresponding to Q1, Q4 of Major issues raised by Reviewer #1.

In the previous version of the manuscript, the authors show the mirror matching of MS/MS spectra between synthetic standards and plasma endogenous peptides (Figure 1F, 5C in the previous version). This represents one of the golden standards for identification of new/unknown peptides in biological samples using mass spectrometry. In the revised version, the authors removed this part and replaced it with a targeted MS using MRM as a previous report by Donohue et al. (PMID: 33360778). However, Donohue et al. used this MRM approach for quantitation of peptide hormones instead of identification. To identify new or unknown peptides, a perfect MS/MS matching between synthetic standards and endogenous peptides is necessary. Besides that, LC retention time matching is required to exclude the possibility of peptide's isomeric difference. We understand that the target peptides' abundance could be too low to obtain high-quality MS/MS spectra using regular flow LC-MS. However, without that, the false identifications of these capped peptides could mislead the scientific community. Therefore, please improve the peptide enrichment and MS detection approaches (perhaps targeted analysis on Orbitrap instruments using nano-flow LC) and obtain high quality MS/MS spectra.

For examples, the two peptides cannot be distinguished by using the authors' MRM approach: ABCD or BACD. The two peptides might have the same molecular weights, similar RT values, and the same b₂, b₃, y₁, y₂ ions, but they are different two peptides. The second example is ABCD or ABCDamide. The amidation only show 1 Dalton mass shift. The two peptides might have the similar RT value and the same b₁, b₂, b₃ ions, and all the y ions only show 1 Dalton difference. The mammalian's proteome is extremely complicated, and mis-matching often occurs. Therefore, mirror matching of MS/MS is necessary.

2. Corresponding to Q4 of Minor issues raised by Reviewer #1.

It is improper that the authors removed the problematic figure and avoided the reviewer's question. It is not difficult to annotate the MS/MS spectrum, because pELELRVAAGRamide is a regular peptide and should result in abundant fragment ions. Why delete that?

Overall, this reviewer suggests further experimental evidences on the MS/MS matching. Without that, the data resource is doubtful.

We have now undertaken extensive efforts to obtain as many MS/MS spectra for mouse and human capped peptides as possible. Now beyond MS/MS spectra for CAP-GDF15 and CAP-TAC1 which we were in the original version of the manuscript, we now have included additional MS/MS spectra which are shown in **Fig. S3** and **Fig. S7**.

As this reviewer notes, the low abundance of these peptides remains a major experimental limitation, which is the reason why we were unable to collect complete MS/MS data for the other capped peptides. However, in those cases, we do have three alternative lines of mass spectrometry evidence for their endogenous presence: 1) co-elution with an authentic peptide

standard, 2) exact mass consistent with the proposed structure, and 3) LC-QQQ evidence for detection based on a specific parent-to-daughter transition. We have therefore now altered the text in two ways to reflect the fact that all capped peptides have the three lines of mass spectrometry evidence for endogenous presence, and that a subset of additional capped peptides, including CAP-GDF15 and CAP-TAC1, also have additional matching MS/MS spectra. First, we soften our language throughout the text. In several places throughout the abstract and Results, we have removed the word “detect” and replaced with “provide mass spectrometry evidence for.” In addition, we have added the following new text in the Discussion to address the future need for MS/MS data, especially for those capped peptides that will be subjected to extensive functional validation: “In addition to the LC-QQQ evidence for all capped peptides, we were successful in obtaining additional complete MS/MS spectra for subset of capped peptides, including the two capped peptides that were subjected to functional validation (CAP-TAC1 and CAP-GDF15). The low abundance of these peptides remains a major experimental limitation for obtaining complete MS/MS spectra, and this is an important area for future work.”

Finally, as pointed out by Reviewer #2, unmodified versions of the peptides described within this manuscript have also been detected in other large-scale peptidomics screens, which provides further independent data for the endogenous presence of capped peptides. We have therefore added the following sentences to the Discussion: “Further independent evidence strengthening the case for the endogenous presence of capped peptides is the fact that other large-scale peptidomics screens have also detected similar sequences to those reported here,²¹ including the VIP-derived peptide QMAVKKLYNSILN (including the amidation)²². One advantage of our approach is the ability to identify very short peptides, compared to more traditional peptidomics screens which usually do not search for peptides shorter than 7-8 amino acids.”

Reviewer #2 (Remarks to the Author):

Overall comments

The authors have done a good job addressing the points I raised, adding new experimental data, analyses and figures/tables. The manuscript represents an advance in our knowledge about peptides by demonstrating the existence in mouse and human blood of many short (3-20) peptides harboring the pyro-Glu / C-term Amidation combination in human and mouse blood. They show bioactivity for some of them and although their primary example (CAP-TAC1) is nearly identical in sequence to several peptides already known to be active (SubstanceP and neurokinins), they have added new data to demonstrate that the pyro-Glu modification leads to increase stability and a different receptor affinity profile. To this reviewer, the GDF15 peptide is still the main biological discovery because it is both truly novel and bioactive.

Specific comments (relating to authors rebuttal)

Add 1) the in-silico analysis

I am glad that my question made the authors realize that they had in fact included GRR in their original analysis, not GKK. But this of course turns attention to the reason for leaving out GKK.

The authors provide the argument that this reviewer should have somehow claimed that GKR and GRR are more prevalent than GKK but I do not recall saying that anywhere in my review. Nor do I think it is necessarily true. A quick search reveals that in mouse, Gastrin-releasing

peptide, Neuromedin-C, Neuromedin-B and Melanocyte-stimulating hormone alpha are all formed from GKK motifs where the G is converted by PAM into the C-terminal amidation. So, PAM certainly works on GKK too.

This is not a critical point, and the revised wording is ok, although it still seems that GKK was left out based on an assumption of not being as relevant as the two other motifs for C-terminal amidation, and that doing so may have left out potential candidate peptides from their in-silico analysis. But I acknowledge and accept that the authors can't redo the whole project to include GKK bearing peptides.

Regarding the question of why the length cut-off was set at 3-20 amino acids, I thank the authors for their honesty in the feedback provided to my comment. Based on that, I suggest that they revise the manuscript further to also let the readers know the real motivation for limiting the length to 20 aa (in the main text when they mention the 3-20 aa search criterium).

The changes made in response to my question is addition of some rather cryptic new wording added about "re-discovering known peptides" plus new comments in the discussion.

I would much prefer if they just said in the main text that they chose 20 aa as the length cut-off because this is the upper limit of their chemical synthesis capabilities and because the only 3 known peptides with the capped peptide characteristics are all shorter than 20 aa. This is a perfectly valid argument for the choices they made in their analysis, and I see no need to "hide" this from the reader.

We have now revised the text in the Results to incorporate this referee's suggestion: "By length, we restricted our search criteria to those peptides 20 amino acids or shorter because this is the upper limit for reliable chemical synthesis of authentic peptide standards; in addition, multiple known peptides with capped characteristics are shorter than this length."

Add 2) Is the number of in-silico identified peptides surprising or not?

The authors have performed the analysis I suggested, and although it is not backed by statistical analyses, it appears – as the authors point out themselves - to show that there is nothing unusual or unexpected about the frequency of peptides with a Q compared to other amino acids. They are no more prevalent in the proteome than would be expected statistically.

This finding would perhaps be worthy of a few comments and reflections, since one would normally expect a higher level of evolutionary conservation from motifs and modifications which play a vital biological role.

In general, I think the authors should consider if they can somehow avoid presenting their in-silico findings as a major surprise in terms of the number of sequences identified (without backing this with evidence), since the evidence suggest that the number of peptides identified in-silico are probably no larger than for any other amino acid than Q.

What the authors have unfortunately done instead in the revised manuscript, is to use the new analysis (of the N-terminal amino acid frequencies) to conclude that their in-silico strategy is worth extending beyond pyro-Gly/C-am.

To me, this is a kind of nonsense argument. If one looks for the occurrence of any given amino acid 3-20 positions upstream of a GKR/GRR motif in a protein, one is bound to find many

examples, and the authors do. But that does not necessarily make this an interesting analysis worth pursuing for peptides with alternative N-terminal residues. The argument also contradicts their own key point, namely that pyro-Glu in combination with C-terminal amidation is something special. Their work indicates that it might well be, so I would omit this comment, as it only adds confusion.

The first new sentence added in their response could also be sharpened to just say that their results demonstrate the existence of many new examples of “capped peptides” detectable in blood. That is interesting in its own right.

I would, however, suggest to leave out all the (unsubstantiated) comments about the number of peptides discovered in-silico or experimentally being much larger than expected or anticipated, unless the authors can provide references to previous work which contains estimates or speculation on the number of pQ/C-am peptides. To my knowledge, there are no references speculating about the potential true number of this particular combination of modifications, and it is therefore unnecessary to claim that this new work finds more than were previously anticipated or expected.

We have now incorporated several of the referee’s comments. First, we have removed all instances of capped peptides being a larger set than previously anticipated. The sentences have now been revised: “Our studies demonstrate that N- and C-terminal capping chemical motif that is present in more endogenous secreted peptides than previously reported.” Second, we have now removed the sentence, “In addition, considering that many other modifications of peptide N-termini have been reported, including N-acetylation and N-formylation, we suspect that the methodology presented here may be more generalizable beyond the set of capped peptides presented here.”

Add 3) Detection of peptides in plasma

The authors have addressed most of the comments, and my only remaining feedback is around the novelty of the identified sequences.

Comparing to PeptideAtlas is a nice addition but it may also be worth considering comparison to other large-scale peptidomics screens. For instance, QMAVKKYLNSILN (derived from VIP) was reported in <https://www.ncbi.nlm.nih.gov/pmc/articles/PMC4857386/> in 2016 (including the amidation), and some of the other sequences reported by the authors as novel were reported to be detectable in whole tissues by Madsen et al. <https://www.nature.com/articles/s41467-022-34031-z>). This strengthens the case for their existence and relevance in-vivo.

The authors could also highlight that their approach has an advantage in the ability to identify very short peptides, compared to MS-based peptidomics screens which usually do not search for peptides shorter than 7-8 aa.

As mentioned earlier, it is not surprising to this review that it is possible to detect sequences which are shorter versions of known bioactive peptides, given all the previous work which has demonstrated the existence of many shorter versions in-vivo. It is thus not the existence of “capped” peptides that is the main story but the fact that at least some of them appear to be both stable and bioactive.

We have now incorporated this reviewer’s comments into the Discussion: “Further independent evidence strengthening the case for the endogenous presence of capped peptides is the fact

that other large-scale peptidomics screens have also detected similar sequences to those reported here,²¹ including the VIP-derived peptide QMAVKKLYNSILN (including the amidation)²². One advantage of our approach is the ability to identify very short peptides, compared to more traditional peptidomics screens which usually do not search for peptides shorter than 7-8 amino acids.”

Add 4) Novelty of the CAP-TAC1 case story

Very nice additions, figures and data which makes it easier for the reader to understand the many highly related sequences while now also demonstrating that the pyro-Glu actually matters to the activity of the peptide. No further suggestions on this from my side.

Add 5) Are “capped peptides” a novel class of peptides?

On this point, the authors have added very nice new data and figures which now provide much better evidence for the pyro-Glu versions being functionally distinct from unmodified variants.

Add 6) The method works for finding bioactive peptides

I very much applaud the changes here which shine additional light on the truly novel bioactive GDF15 peptide discovered by the authors. The fact that CAP-TAC1 is active against the same receptors as Substance P is not surprising given that CAP-TAC1 is a sub-sequence of Substance P with a pyro-Glu modification. But demonstrating that the novel GDF15 peptide also lowers appetite is a significant discovery because this peptide is new and non-overlapping with the well known GDF15 peptide. The observation that the RR site which is cleaved to form the classical GDF15 (MIC1) peptide has an upstream G support the case for conversion of this glycine to an amidation that the authors detect.

Add 7) Formation of pyro-Glu

No further comments on this part, following the new additions which provide evidence against the idea that the pyro-Glu modifications could be experimental artifacts, and new data demonstrating differences between Glu and pyro-Glu versions of the same peptide.

Reviewer #3 (Remarks to the Author):

The authors have taken into account my previous comments, and I am happy with most of the adjustments.

I however still have an issue with the administration of the GDF15 peptide to the mouse. The authors dose a mouse at 50 mg/kg, therefore taking a conservative value of a mouse being 30g, approximately 1.5 mg of the peptide was dosed in one animal.

According to figure S4A, this caused an increase of the endogenous peptide level from approximately 1 nM to a maximum of 4 nM. A concentration of 4 nM is equivalent to around 5 ng/mL for the endogenous GDF15 peptide, therefore the blood concentration was raised from around 1 to 5 ng/mL after the dosing of 1.5 mg of peptide.

Whilst there isn't enough data to work out the area under curve of the administration, this does suggest that the bioavailability of the peptide via an I.P route administration is incredibly poor.

For example, comparing the data from Liraglutide administration

(<https://doi.org/10.2337/diacare.25.8.1398>) in humans at 5 µg/kg (10,000 times lower dosing than this study), equating to approximately 0.4 mg peptide dosage to a human based on 80kg

weight (3x less overall peptide dosed than this study – but to a human). This liraglutide administration to humans returned a similar CMax of 4.5 nM, and with a bioavailability of 55%. See data in table 1.

We have now added additional text to the Discussion of the 50 mg/kg dosing for CAP-GDF15, and the need for further characterization of the pharmacokinetics and pharmacodynamics of this molecule: “Lastly, our initial studies of CAP-GDF15 shown here use a relatively high dose of 50 mg/kg. In the future, it will be important to establish the full dose-response and pharmacokinetic/pharmacodynamic profile of CAP-GDF15.”